# Composition and Distribution of Plankton Communities in the Atlantic Sector of the Southern Ocean

Valentina V. Kasyan [1,*], Dmitrii G. Bitiutskii [2,3], Aleksej V. Mishin [4], Oleg A. Zuev [4], Svetlana A. Murzina [3], Philipp V. Sapozhnikov [4], Olga Yu. Kalinina [4], Vitaly L. Syomin [4], Glafira D. Kolbasova [5], Viktor P. Voronin [3], Elena S. Chudinovskikh [2] and Alexei M. Orlov [4,6,7]

[1] Laboratory of Systematics and Morphology, A.V. Zhirmunsky National Scientific Center of Marine Biology, Far Eastern Branch, Russian Academy of Sciences (NSCMB FEB RAS), 690041 Vladivostok, Russia

[2] Sector of the World Ocean, Azov-Black Sea Branch of the Russian Federal Research Institute of Fisheries and Oceanography ("AzNIIRKH"), 344002 Rostov-on-Don, Russia

[3] Environmental Biochemistry Laboratory, Institute of Biology of the Karelian Research Centre of the Russian Academy of Science (IB KarRC RAS), 185910 Petrozavodsk, Russia

[4] Shirshov Institute of Oceanology, Russian Academy of Sciences (IO RAS), 117997 Moscow, Russia

[5] N.A. Pertsov White Sea Biological Station, Faculty of Biology, M.V. Lomonosov Moscow State University (WSBS MSU), 119234 Moscow, Russia

[6] Laboratory of Behavior of Lower Vertebrates, A.N. Severtsov Institute of Ecology and Evolution, Russian Academy of Sciences (IPEE RAS), 119071 Moscow, Russia

[7] Department of Ichthyology and Hydrobiology, Biological Institute, Tomsk State University (TSU), 634050 Tomsk, Russia

* Correspondence: valentina-k@yandex.ru

**Abstract:** In recent decades, the waters off the Antarctic Peninsula and surrounding region have undergone a significant transformation due to global climate change affecting the structure and distribution of pelagic fauna. Here, we present the results of our study on the taxonomic composition and quantitative distribution of plankton communities in Bransfield Strait, Antarctic Sound, the Powell Basin of the Weddell Sea, and the waters off the Antarctic Peninsula and South Orkney Islands during the austral summer of 2022. A slight warming of the Transitional Zonal Water with Weddell Sea influence (TWW) and an increase in its distribution area was detected. Among the pelagic communities, three groups were found to be the most abundant: copepods *Calanoides acutus*, *Metridia gerlachei*, and *Oithona* spp., salpa *Salpa thompsoni*, and Antarctic krill *Euphausia superba*. Euphausiids were found in cases of low abundance, species diversity, and biomass. In the studied region, an increase in the amount of the salpa *S. thompsoni* and the euphausiid *Thysanoessa macrura* and the expansion of their distribution area were observed. Significant structural shifts in phytoplankton communities manifested themselves in changes in the structure of the Antarctic krill forage base. The composition and distribution of pelagic fauna is affected by a combination of environmental abiotic factors, of which water temperature is the main one. The obtained results have allowed us to assume that a further increase in ocean temperature may lead to a reduction in the number and size of the Antarctic krill population and its successive replacement by salps and other euphausiids that are more resistant to temperature fluctuations and water desalination.

**Keywords:** thermohaline structure; phytoplankton; macro- and mesozooplankton; ichthyoplankton; abundance; biomass; distribution; Southern Ocean

## 1. Introduction

In recent decades, the waters off the Antarctic Peninsula and the surrounding region have undergone a significant transformation due to global climate change [1]. The most noticeable consequence of this process is a huge reduction in the area of ice cover and the proportion of old sea ice [2]. Since the middle of the 20th century, a significant warming has been observed in the Southern Ocean: the temperature of its upper layers

westward of the Antarctic Peninsula has increased by more than 1 °C since 1955 [3]. The degradation of the ice cover entails changes in hydrophysical conditions, the duration of the production period, and the structure and distribution of plankton communities. The Antarctic pelagic ecosystem is now undoubtedly in a state of transformation [4–8]. Phyto-, zoo- and, ichthyoplankton play an important role in the functioning of ecosystems in this part of the World Ocean [9,10]. Most planktonic organisms are characterized by short lifecycles and are the first to react to climate changes [11–13]. The Antarctic krill *Euphausia superba* (hereafter referred to as krill) is one of the largest and most abundant species of crustaceans in the Southern Ocean [14] and plays a key role in the trophic chains of this region [15]. It is, therefore, quite natural that the role of euphausiids, especially Antarctic krill, in maintaining balance in the circulation of organic matter and energy in the changing Southern Ocean has been actively discussed recently [16]. There is a steady trend towards a reduction in the population of Antarctic krill and its sequential replacement with salps [17–19]. Current climate changes are reflected in a peak increase in the number of salps on the southern boundary of its range and the shift of this boundary from 60° S to 65° S. As a result, the ranges of salpa and Antarctic krill overlap, which leads to their direct competition for habitats and forage [20]. In addition, it is noted that this phenomenon (the overlap between the habitats of the main macrozooplankton species) can be both a part of a natural systemic mechanism [21–23] or may cause negative consequences for Antarctic krill populations and the Southern Ocean ecosystem as a whole [18,24,25]. From 1993 to 2004, periods showing the abundance of salps were recorded to the west of the Antarctic Peninsula [26], which normally indicates warm water masses [27], which transfer low to moderate levels of chlorophyll a [28,29]. Unlike krill, salps do not depend on ice cover for their development, nor do they need the associated ice algae, which are the main food source for krill larvae and juveniles [30].

All these changes need to be monitored since Antarctic krill is a valuable and commercially exploited species, and thorough scientific analysis is required for catch quota allocation. The waters around the Antarctic Peninsula, the South Orkney Islands, and most of the Weddell Sea have been designated by the Commission for the Conservation of Antarctic Marine Living Resources (CCAMLR) as Marine Protected Areas (MPAs)—the existing ones and those which are currently being developed or discussed [31,32]. Therefore, regular research activities in this region are critically important for obtaining reliable scientific information on the composition and structure of the communities and their dynamics [33]. Such studies are of particular importance when considering issues related to the creation of MPAs, monitoring the state of biota both within and outside the MPAs, as well as when making related management decisions [34]. The results of the study are of indisputable practical importance for monitoring the current state of Antarctic ecosystems and their biological resources in the face of changing environmental factors, including current trends in climate change, which is the primary basis for considering issues and problematic situations related to MPAs and their status in this region.

## 2. Materials and Methods

### 2.1. Oceanographic Measurements

Data were collected during the 87th cruise of the R/V *Akademik Mstislav Keldysh* in January–February 2022. Research areas included the Bransfield Straits, the Antarctic Sound, the Powell Basin of the Weddell Sea, the waters off the Antarctic Peninsula, and the South Orkney Islands (Figure 1, Table S1).

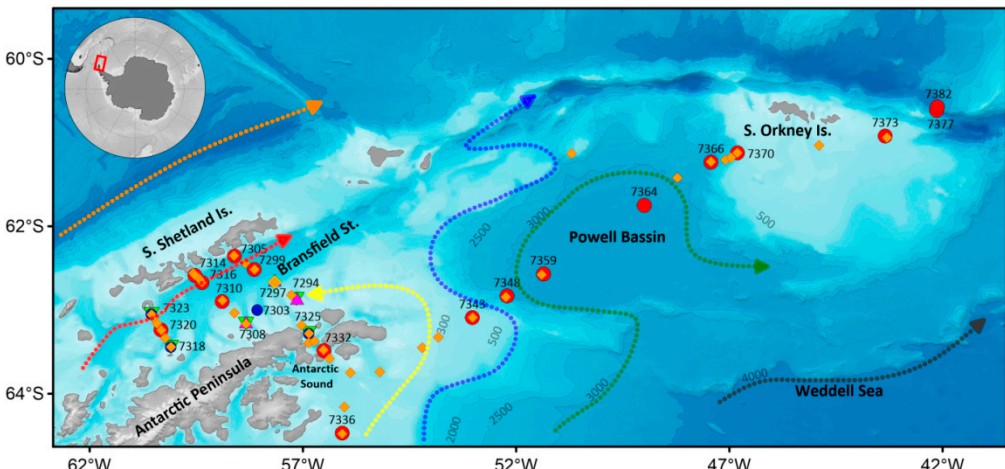

**Figure 1.** The sampling stations and the main currents off the Antarctic Peninsula. Numbers represent names of complex stations (except for the station 7303). Dotted lines represent currents, according to [35,36]. Orange line—the Antarctic Circumpolar Current (ACC), red line—the Bransfield Current (BC), yellow line—the Antarctic Coastal Current (ACoC), blue line—the Antarctic Shelf Front (ASF), green line—the Weddell Front (WF), grey line—Weddell Deep Water (WDW). Plankton nets: red circle—Double Square Net (DSN) and Multinet integrated, blue circle —DSN, orange diamond—Apshtein, pink triangle—Bongo, green triangle—WP-2.

The oceanographic measurements at stations were performed using a (Lowered Acoustic Doppler Current Profiler (LADCP) and Conductivity, Temperature, Depth (CTD) profilers mounted on a General Oceanics GO1018 rosette water sampler. The Idronaut OCEAN SEVEN 320Plus probe (Idronaut, Italy) was equipped with a high-precision, temperature-compensated pressure sensor (PA–10X) with an accuracy of 0.01% and a resolution of 0.002% for the full measurement range (0–100 MPa) and two redundant temperature sensors with a measurement range between −5 °C and 45 °C and an initial accuracy of 0.001 °C and resolution of 0.0001 °C. Two redundant conductivity sensors had a measurement range of 0 to 7 S/m, an initial accuracy of 0.0001 S/m, and a resolution of 0.00001 S/m. The currents were measured with a TRDI Workhouse Monitor (Lowered Acoustic Doppler Current Profiler, LADCP, Teledyne Technologies Inc., Southend Oaks, CA, USA) submersible acoustic Doppler profiler with a frequency of 300 kHz, paired with a shipborne acoustic Doppler current profiler (LADCP) TRDI Ocean Surveyor-75 (Teledyne Technologies Inc., Southend Oaks, CA, USA) with a frequency of 75 kHz. The obtained data were processed using the LDEO Software ver. IX.10 [37]. The final measurement accuracy was 3–4 cm/s for bottom layers up to 1–2 cm/s due to the bottom track. Additionally, tidal forces were taken into account by using the software described in [38]. In situ chlorophyll a (Chl a) profiles were measured and integrated over the depth of the water column (100 m to surface) with a calibrated fluorometer (Seapoint Sensors, Inc., Exeter, NH, USA) mounted on the CTD.

*2.2. Biological Sampling*

Samples of phytoplankton were taken at 41 stations using the medium Apshtein net (38 µm mesh; 0.36 m mouth diameter) (Figure 1). The depth range of 0–50 m was sampled; at some stations, a 50–160-m tow was added.

Mesozooplankton samples were obtained at 23 complex stations by a Multinet opening/closing net system (0.25 m$^2$ aperture) equipped with five nets with a 150-µm mesh-size [39] by performing vertical tows from 500 m to 300, 300–200, 200–100, 100–50, and 50–0 m. At the non-deep-sea stations, sampling was performed using the vertical tows of a WP-2 net with a 150-µm mesh-size [40] from 200 m to the surface or from near-bottom to the surface at depths <200 m (Figure 1). Samples were stored in plastic 100-mL vials in 4% buffered formaldehyde until future analysis.

Macrozooplankton and ichthyoplankton samples were obtained at 23 complex stations (+stn. 7303) using a Bongo net (505 μm mesh, 0.6 m mouth diameter) [41] by performing oblique tows from 200 m to the surface and by using a pelagic double square micronekton net (DSN) (505 μm mesh, 1.0 m$^2$ inlet area) [42] (Figure 1) equipped with a pterygoid deepener weighing 24 kg (Hydrobios, Altenholz, Germany), as well as by performing oblique tows from 600 m to the surface at an average speed of 1.5 knots. DNS and Bongo nets were equipped with a water flow counter (Hydrobios, Altenholz, Germany).

### 2.3. Taxonomic Identification

The current taxonomic names of all species were verified according to the World Register of Marine Species [43]. Phytoplankton samples were examined live on board the vessel using a Carl Zeiss Primo Star microscope (Carl Zeiss, Jena, Germany), with a parallel photographic recording of all the examined species. Photographs were taken by a ToupCam 5.1 MP digital camera (ToupTek, Hangzhou, China). The species' abundance and the stages of their lifecycle were established at that stage. After viewing, the preparations were washed back into the samples. Then, the samples were fixed with a 40% ethanol solution, precipitated, concentrated, and stored in a dark box until they arrived at the onshore laboratory. There, the concentrated material was thoroughly mixed and viewed using Leica DMLS and Leica DM2500 microscopes (Leica, Wetzlar, Germany) as well as photographed by the digital cameras of BlackView smartphones (16 MP, Blackview, Shenzhen, China) and the built-in microscope camera Leica DM2500 (5 MP, Leica, Wetzlar, Germany). Detailed quantitative analyses were carried out, species after species, taking into account the stage of the lifecycle: spores and cysts were counted separately from vegetative cells. To identify species, up-to-date guidebooks were used, as well as specialized publications on the algal flora of the Weddell Sea and the Atlantic [44–51].

Mesozooplankton were identified to the lowest possible taxonomic level in a Bogorov chamber using the stereo microscopes SZX7 and SZ51 (Olympus, Tokyo, Japan) and a compound microscope Leica DM2500 (Leica, Wetzlar, Germany) according to available keys [52–59]. Large organisms (total length ≥ 2 cm) were first washed, separated, and counted. Copepods and krill larvae, as a dominant group of mesozooplankton, and holopelagic polychaetes, as a less abundant but significant group of mesozooplankton, were subjected to a more detailed taxonomical processing. Other mesozooplanktonic groups, e.g., hyperiids, chaetognaths, and pteropods, were occasionally found in the study area and were therefore not described; however, their abundance was evaluated and added to the total abundance.

Euphausiids and salps, as very abundant groups of microzooplankton, were processed immediately in accordance with the generally accepted methods and recommendations [60,61]. The entire sample of salps was fixed with a 4–6% formalin solution. Taxonomic affiliation was determined under the Ulab WF20X stereo microscope (ULAB Scientific Instruments Co., Ltd, Nanjing, China) at magnification ×30 or by using the digital USB microscope ADSM 301 (Shenzhen Andonstar Technology Co., Ltd., Shenzhen, China) at a digital scaling up to ×4 using the identification keys [62,63].

Ichthyoplankton samples were stored in a plastic container in a 2% formaldehyde solution or in 96% ethanol in case further genetic studies were planned. The identification of larvae was carried out to the species level wherever possible using some previously published keys [64–67].

### 2.4. Data Analysis

All calculations and statistical tests were performed using PRIMER ver. 6 [68] with the PERMANOVA+ 1.0.6 extension [69]. Maps of the sampled areas and the spatial distribution of the pelagic communities were drawn in the SURFER ver. 20 (Golden Software, Golden, CO, USA) package.

### 2.4.1. Phytoplankton Community Analysis

To assess the similarities and differences between communities, the Sorensen index (qualitative features) and the Bray–Curtis similarity index (quantitative structures) were used. Further analysis was carried out using hierarchical clustering and non-metric multidimensional scaling (nMDS). Floristic and coenotic groups were identified using the similarity procedure (ANOSIM), while sets of species characterizing certain groups were identified using the similarity percentage analysis (SIMPER) [70]. Here, we did not present similarity dendrograms of communities at sampling stations; instead, we chose to show the maps of the groups' distribution over water areas. There were two reasons for this: (1) the dendrograms for 41 stations are cumbersome and do not look informative in a compact illustration size; (2) only 33 communities from the 41 stations were included in the groupings identified with a high degree of reliability; the rest of the dendrograms are not informative. To designate community groupings, a sequence of Latin letters was used, adding an "f" for floristic ones and a "c" for coenotic ones.

### 2.4.2. Mesozooplankton Community Analysis

Similarities between stations were assessed using the Bray–Curtis similarity index [71] for the quantitative data (densities of organisms of each taxonomic group in samples). Species compositions were compared using the taxonomic dissimilarity Gamma+ [72]. Ordination was performed using nMDS and principal coordinates analyses (PCO) [73]. Reliability of grouping was tested using the ANOSIM method. Inter- and intra-group similarities were assessed using the permutational analyses of variance (PERMANOVA) pairwise tests [69]. Factor analysis was performed using the distance-based linear models (DistLM) routine [69]. Diversity was assessed using the Shannon–Weaver index (H'). The ice-free time at each station was estimated using the database "Circumpolar Ice Maps of the Southern Ocean", available on the Arctic and Antarctic Research Institute website (AARI-NIC-NMI pilot project on integrated sea ice analysis for Antarctic waters http://ice.aari.aq/ (accessed on 11 May 2022).

### 2.4.3. Macrozooplankton Community Analysis

The density of salps and euphausiids aggregations was calculated as the ratio of the number of these organisms in the catch per 1000 m$^3$ of filtered water [74]. The similarity between stations based on quantitative data (density of organisms in a sample) was assessed using the Bray–Curtis index [71]. To assess the reliability of clustering, the SIMPROF permutation test (number of permutations 999, $p = 0.05$) was performed [68].

### 2.4.4. Ichthyoplankton Community Analysis

The total number of fish at the early stages of development in a sample was recalculated into a relative one (ind/m$^2$) for a more precise comparison of the stations according to the formula:

$$N = (n \times H)/V$$

where N is the relative density, n—number of larvae in a sample, H—towing range, m, V—volume of filtered water, m$^3$ [75]. Statistical data processing was based on the similarity matrix with the Bray–Curtis index as a similarity measure [71–76]. We used the number of individual fish species in the sample, recalculated per 1 m$^2$ of water surface [75] and expressed as a percentage of the total number of larvae in the sample as the input data. Before calculating the matrix, in order to reduce the influence of highly dominant species, the original data were square rooting transformed. Hierarchical clustering was performed using the unweighted pairwise group average (UPGMA) method. To assess the reliability of clustering, the SIMPROF permutation test (number of repeats 999, $p = 0.05$) was performed [68].

## 3. Results

### 3.1. Oceanographic Characteristics

The distribution of thermohaline properties and bathymetry has allowed us to identify three sub-regions within the research region: the Bransfield Strait, the Antarctic Sound with southern stations, and the Powell Basin, including the area northeast of the S. Orkney Islands (Figure 2). The maximum temperatures in upper 300 m were observed to the northeast of the S. Orkney Islands. Other warmer waters were present at the surface in the northwestern Bransfield Strait and in most of the Powell Basin (Figure 2A). An eastward increase in temperature was observed. Positive potential temperature was traced throughout the depth along the South Shetland Islands. Local minimums of potential temperature were registered to the south of the Antarctic Sound and on the southwestern slope of the Powell Basin. High salinity was observed in the northeastern part of the Bransfield Strait, and low salinity was detected in the waters near the S. Shetland Islands and northeast of the S. Orkney Islands (Figure 2B).

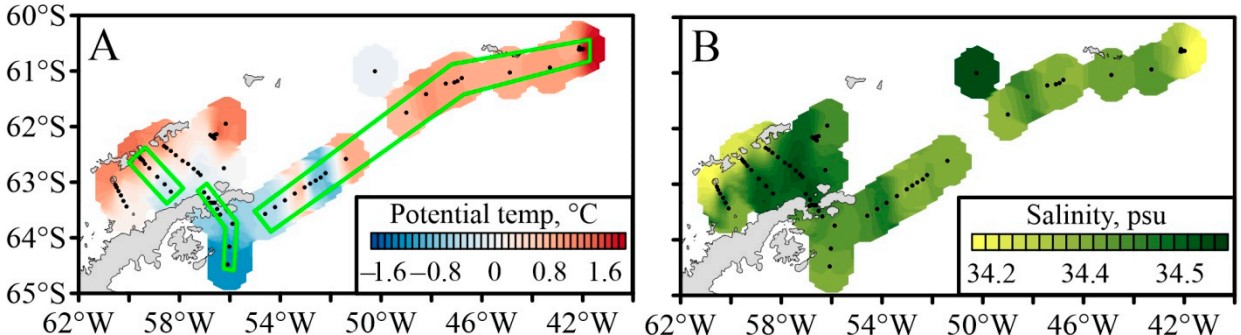

**Figure 2.** Spatial distribution of potential temperature (**A**) and salinity (**B**) in the study area at depths of 0–50 m. Stations are marked with black dots; sections considered below are colored in green.

Bransfield Strait (Figures 3 and 4) was filled with cold and saline Transitional Zonal Water with Weddell Sea influence (TWW), spreading from north to south. Transitional Zonal Water with Bellingshausen Sea influence (TBW) was propagated to the northeast along the South Shetland Islands in the form of a narrow high-velocity jet called the Bransfield Current (BC) and only reached the middle of the Strait in the upper 50 m. Modified Circumpolar Deep Water (mCDW) was observed at a depth of 300 m. The maximum potential temperature of 1.70 °C and a minimum salinity of 34.12 psu were recorded in the surface layer at stn. 7313. The minimum potential temperature of −0.80 °C was observed at a depth of 300 m in the middle of the Strait. The maximum salinity of 34.68 psu corresponded with the core of mCDW and was observed at stations 7314 and 7317. The waters near the S. Shetland Islands (Figure 4, stn. 7314) were stratified more by salinity than by potential temperature. The waters of the central part of the Strait were stratified mainly by the potential temperature, while the variations in salinity did not exceed 0.10 psu (Figure 4, stn. 7310). All three water masses were well defined by the 0 °C isotherm.

In the Antarctic Sound (Figures 3 and 4), potential temperature and salinity decreased from north to south. The northern shallow part of the Strait was filled with the waters of the Bransfield Strait characterized by higher potential temperatures, reaching 0.09 °C in the upper 60 m layer and a higher salinity level of up to 34.55 psu in the bottom layer. Pronounced upwelling was observed in the middle of the Strait on the dump of the depths, and the −0.60 °C isoterm was located at a depth < 30 m. The deep-water part of the Sound was subject to the influence of Weddell Sea waters, as well as active ice melting, which was stronger than in previous years. Significant freshening was observed in the upper 50 m layer, where the minimum salinity was 34.29 psu. South of the Antarctic Sound, the potential temperature was <−0.60 °C throughout the depths, except for the near-surface layer. Station 7336 was located on the boundary of packed ice and was characterized by the minimal salinity (33.55 psu) and positive potential temperature (0.12 °C) in the near-surface

layer. The potential temperature observed at a depth of 200 m (−1.80 °C) was close to the freezing point.

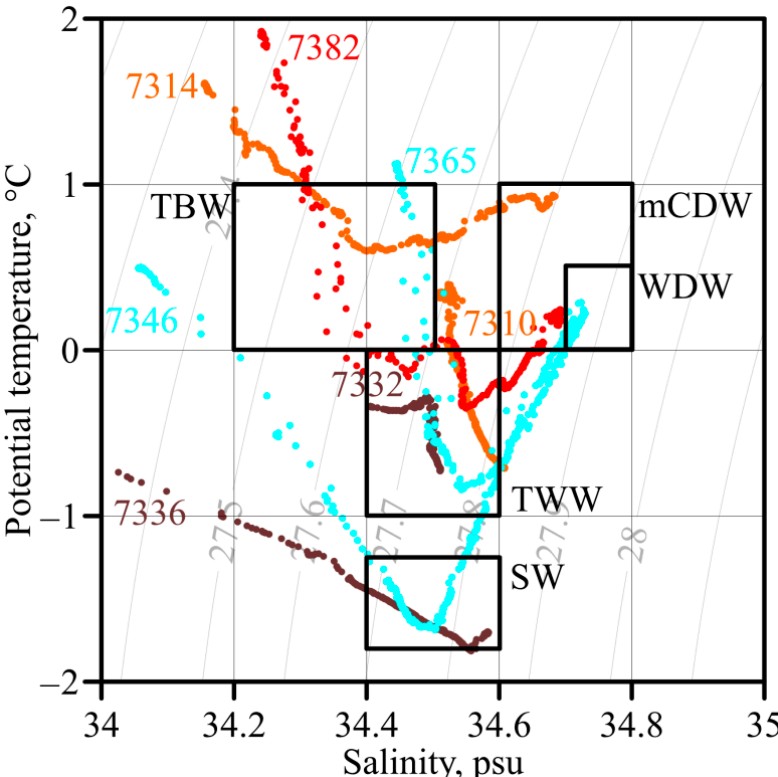

**Figure 3.** θ, S-curves of individual stations for each area: orange—the Bransfield Strait, brown—the Antarctic Sound, cyan—the Powell Basin, red—South Orkney Islands shelf. Black boxes: Transitional Zonal Water with Bellingshausen Sea influence (TBW), Transitional Zonal Water with Weddell Sea influence (TWW), modified Circumpolar Deep Water (mCDW), Warm Deep Water (WDW), Shelf Water (SW). The gray lines indicate the potential density at sea surface.

In the Powell Basin, three water masses were clearly designated (Figures 3 and 4): Antarctic Surface Water (AASW), Cold Intermediate Layer (CIL), and Warm Deep Water (WDW). The AASW occupies the upper layer of the water column with a thickness varying between 20 and 100 m, depending on the station considered. In this layer, the potential temperature increased from south to north from 0.50 °C to 2 °C, while the salinity distribution showed some local minimums associated with ice melting (34.40–34.50 psu). Below, there was a CIL or Winter Water (WW) caused by the winter convection. Its thickness varied from 50 to 150 m, and the AASW located above and WDW below served as conditional boundaries. The minimum potential temperature of −1.6 °C was observed at the slope stations. WDW was registered in the deeper part of the Powell Basin, below a depth of 200 m, and was well defined by the 0 °C isotherm and 34.70 psu isohaline. At the shelf stations, the potential temperature decreased from the surface to the bottom from −0.32 °C to −1.10 °C, and the salinity decreased from 34.23 psu to 34.55 psu, while WDW was not recorded here.

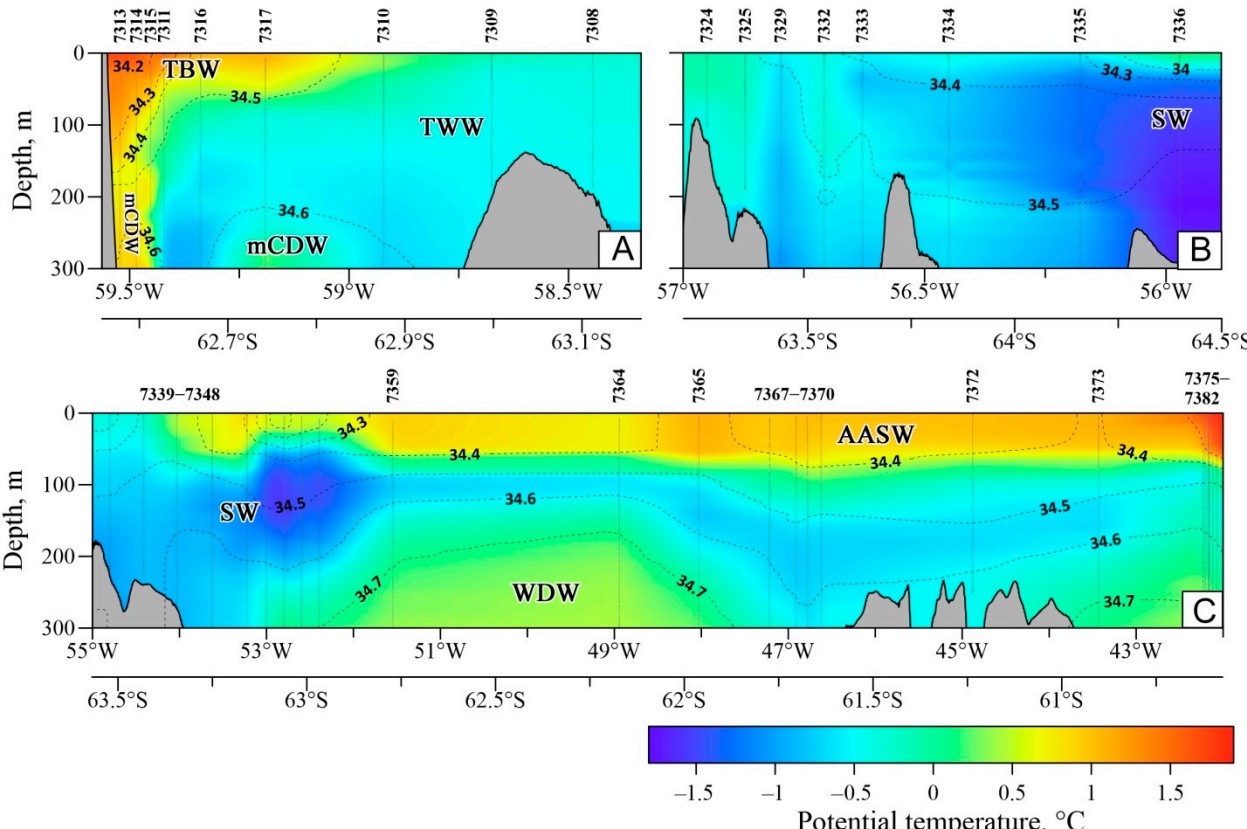

**Figure 4.** Vertical distribution of potential temperature (°C, color) and salinity (psu, dotted line) on the sections: (**A**)—the Bransfield Strait, (**B**)—the Antarctic Sound, (**C**)—the Powell Basin and area northeast of the South Orkney Islands. Station numbers are located at the top of the diagram, the geographic location of the station is shown as a vertical line. The bottom topography is taken from the GEBCO2021 database https://www.gebco.net/data_and_products/gridded_bathymetry_data/ gebco_2021/ (accessed on 19 May 2022).

The thermohaline structure of the waters northeast of the S. Orkney Islands (Figure 3) was found to be similar to that of the deep waters of the Powell Basin. The main feature in this area was the presence of a warmer and fresher surface layer (Figure 4). Potential temperature and salinity reached 2 °C and 34.22 psu, respectively. At a depth of 50–80 m, a sharp seasonal pycnocline was recorded, in which the potential temperature and salinity gradients reached 0.50 °C and 0.06 psu per 10 m, respectively.

A two-jet system was observed in the Bransfield Strait: a weak, wide continuation of the Antarctic Coastal Current (ACoC) to the southwest and a narrow, high-velocity Bransfield Current (BC) along the South Shetland Islands to the northeast. Variable currents were observed in the Antarctic Sound, strongly depending on the time of observation. To the south of the Sound, a high-speed flow was observed, which corresponded to the ACoC, and was localized on the shelf.

Mean values of Chl *a* concentrations were calculated for the upper 100 m of the water column, which was the estimated average depth of the euphotic zone. The concentrations were highest south of the Antarctic Sound and southwest of the South Orkney Islands (>2.4 mg m$^{-3}$) and lowest throughout the waters of the Powell Basin (Figure 5A). Intermediate concentrations with an even gradation were observed in the Bransfield Strait and southeast of the South Orkney Islands. The maximum values' Chl *a* concentrations in the upper 30 m of the water column were recorded south of the Antarctic Sound (stn. 7336) (Figure 5B).

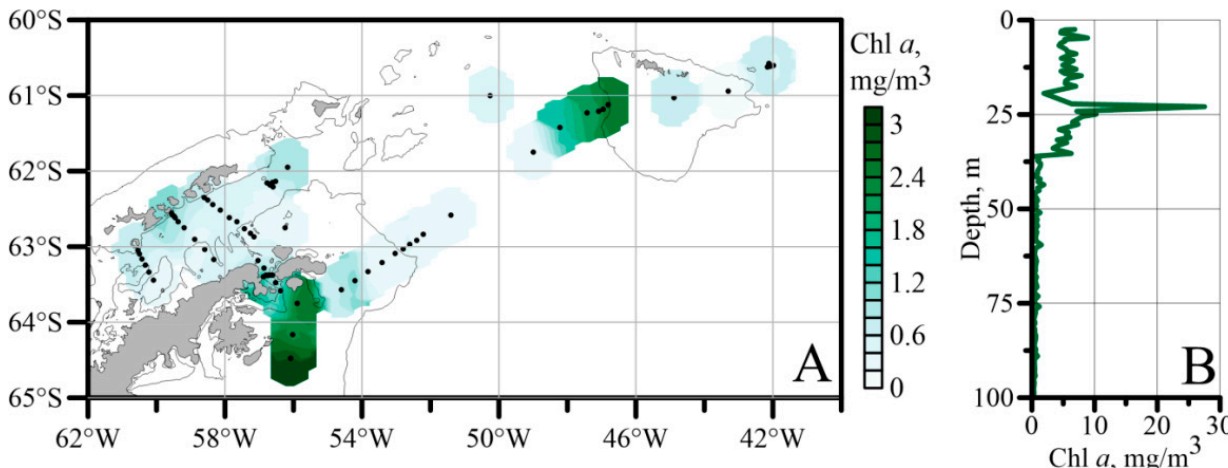

**Figure 5.** Spatial (**A**) and vertical ((**B**), stn. 7336) distribution of chlorophyll *a* concentrations (mg/m$^3$) in the upper 100 m of the water column. Stations are marked with black dots.

*3.2. Phytoplankton*

Results of the analysis of the phytoplankton communities revealed a rich and diverse microphytic flora. In total, in the Bransfield Straits, the Antarctic Sound and the northern Weddell Sea (Powell Basin and the waters of the South Orkney Islands), 109 species and subspecies of microphytes were found. Most of them had a set of adaptations for soaring in the pelagial at the vegetative stages of their lifecycle. The species list was dominated by diatoms (87 species and subspecies, or 79.8% of the total flora). They were followed by dinoflagellates (19 or 17.4%). Haptophytes (*Phaeocystis antarctica*), silicoflagellates (*Dictiocha speculum*), and cryptophytes (*Geminigera criophila*) were represented by one species each. Among the diatoms, the genera *Chaetoceros* (15 species and subspecies or 13.7% of the total flora), *Thalassiosira* (12 or 11.0%, respectively), *Fragilariopsis* (8 or 7.3%, respectively), and *Pseudo-nitzschia* (6.0 or 5.5%, respectively) were especially rich in species, while, among dinoflagellates, it was *Protoperidinium* (12 or 11.0%, respectively).

Some species were present in the communities at different stages of their lifecycle. Among the diatoms, 12 species were noted (*Chaetoceros neglectus, C. socialis, C. gelidus, C. tortissimus, Odontella weissflogii, Porosira glacialis, Stellarima microtrias, Thalassiosira diporocyclus, T. gravida, T. tumida, T. rotula,* and *Thalassiosira* sp. 1), represented not only by vegetative cells, but also by resting spores. For *Corethron pennatum*, the formation of gametes was recorded in many areas. The dinoflagellate *Polarella glacialis* was found only south of the Antarctic Sound (stn. 7336), mainly in the form of cysts. As for *P. antarctica*, they were found in communities in three forms: epiphytic attached; early stages of colony development, leading a predominantly attached lifestyle on diatom hets; and free-floating colonies.

None of the species were found everywhere. The widest distribution was noted for six species at the vegetative stage of development (*Proboscia inermis, Fragilariopsis sublinearis, P. antarctica* (in epiphytic form), *P. glacialis, O. weissflogii,* and *C. pennatum*), and these were observed in over the half of the studied locations. Species such as *Halamphora* sp. 1 (in pelagic aggregates formed on the basis of pellets), *Phalacroma equalanti, Protoperidinium latistriatum, P. bipatens, P. depressum, P. variegatum,* as well as spores of *P. glacialis* and *Thalassiosira* sp. 1, were found only at depths below 50 m.

The floristic groupings that were identified by the similarity of the qualitative composition of the communities and the groups of cenoses identified by the similarity of the quantitative structure were not always confined to certain areas. Thus, four stable floristic groupings were identified (Figure 6). Among them, the «Af» grouping (mean within-group similarity—or average similarity—hereinafter AS—62.4%) was registered in the Antarctic Sound and to the south of it and in the Bransfield Strait off the Antarctic Peninsula. The generatrixes (unifying) species of this grouping were *Fragilaria islandica, Trichotoxon reinboldii,*

*P. inermis*, *Eucampia antarctica*, *F. sublinearis*, *F. cylindrus*, *O. weissflogii*, *Chaetoceros tortissimus*, *Proboscia truncate*, and *C. pennatum* at the vegetative stage and the spores of *C. socialis* and *C. gelidus*. The «Bf» grouping (AS 36.8%) was distributed in the shelf waters off the South Orkney Islands and in the Bransfield Strait. For it, the generatrixes were *Rhizosolenia antennata* f. *semispina*, *Fragilariopsis kerguelensis*, and *C. pennatum*. The «Cf» grouping, which had an AS of 37.6%, was characterized by a relatively compact distribution in the Bransfield Strait near the South Shetland Islands and was also recorded in the Powell Basin. The unifying species here were *C. pennatum*, *P. glacialis*, and *O. weissflogii*. Grouping «Df», at AS 53.3%, was distributed in the northern part of the eastern section across the Bransfield Strait and was also encountered in the western part of the Powell Basin. *P. antarctica* at the epiphytic stage and *C. pennatum* were the most significant species for «Df».

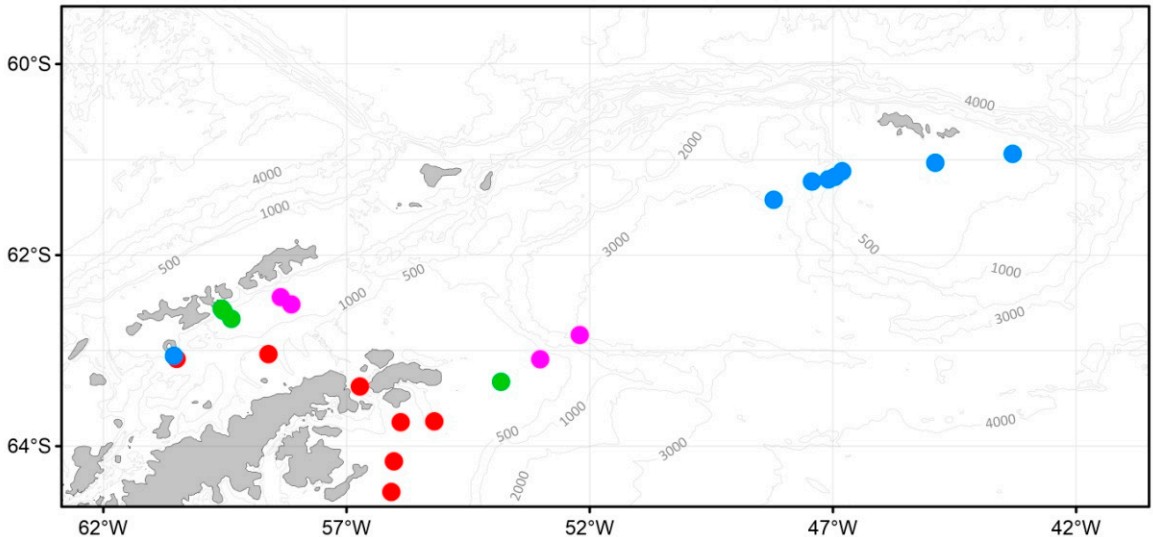

**Figure 6.** Distribution of floristic groupings of phytoplankton. Groupings: red circle—«Af», blue—«Bf», green—«Cf», pink—«Df». See Section 2.4.1 for grouping abbreviations.

We also identified six cenotic groupings (or stable groups of cenoses) (Figure 7). Of these, the «Ac» group (AS 69.2%) was distributed mainly in the northern Bransfield Strait; however, it was also found in its central part. This grouping was characterized, in decreasing order of importance for within-group similarity, by *C. pennatum* and young colonies of *P. antarctica*. Grouping «Bc» (AS 57.1%) was distributed in the central part and south of the Antarctic Sound. This grouping was characterized by a large number of species (14, of which 10 were colonial forms), among which the mature colonies of *P. antarctica* and chainlike colonies of the large cells of *O. weissflogii* were especially significant. In addition to the species at the vegetative stage, this grouping was represented by the spores of two *Chaetoceros* species. The group of cenoses «Cc» (AS 43.3%) was recorded in the southern part of the Antarctic Sound and in the Bransfield Strait; colonial forms of diatoms prevailed everywhere. Ten species characterized this group, among which the most significant were the vegetative colonies of *C. gelidus* and *O. weissflogii*. Spores of *C. gelidus* also played an important role in the structure of these cenoses—they were third in terms of abundance. The «Dc» grouping (AS 23.4%) occurred to the west and southeast of the South Orkney Islands. Among the eight species characteristic of these cenoses, the most important were the vegetative *Actinocyclus* spp., *Asteromphalus parvulus*, and *F. kerguelensis*; they were accompanied by other colonial forms of diatoms (four species). Grouping «Ec» (AS 43.4%) was identified in the southern parts of the central and eastern sections across the Bransfield Strait and in the northern part of the Antarctic Sound. These cenoses consisted of 12 species, where the majority belonged to the vegetative *O. weissflogii* and *C. socialis*. Colonial diatoms also prevailed in the structure of these communities (nine out of twelve species). The «Fc» grouping (AS 35.6%) was a set of relatively species-poor communities, the distribution

of which was not clearly tied to specific areas. These communities were recorded in the Bransfield Strait, the Powell Basin, and south of the South Orkney Islands. These cenoses included three species only, mostly *C. pennatum* and *C. criophilus*.

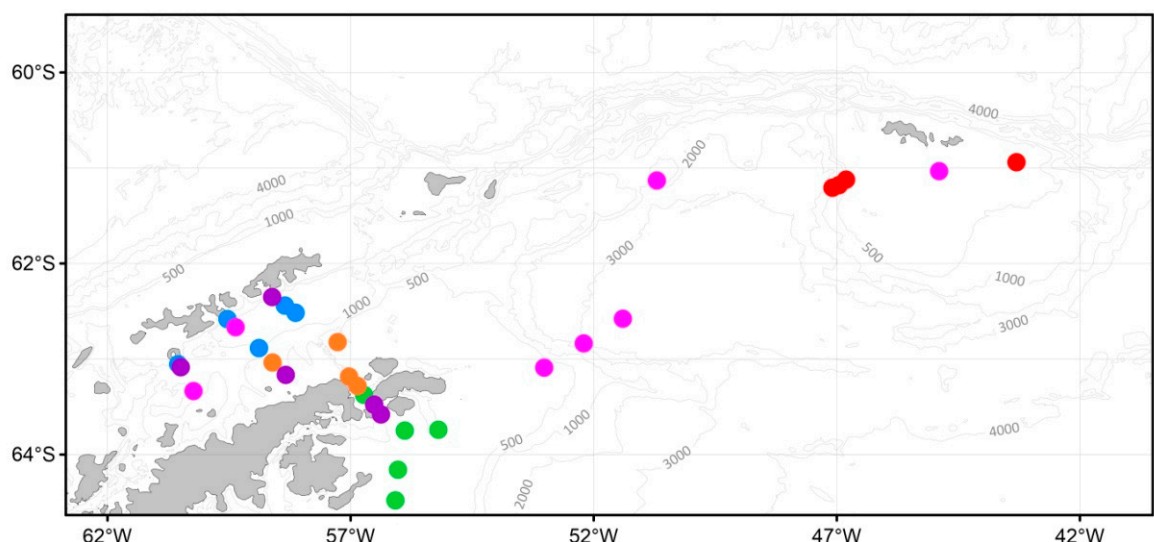

**Figure 7.** Distribution of the coenotic groupings of phytoplankton. Groupings: blue circle—«Ac», green—«Bc», violet—«Cc», red—«Dc», orange—«Ec», pink—«Fc». See Section 2.4.1 for grouping abbreviations.

### 3.3. Mesozooplankton

In the present study, 42 mesozooplankton taxa were identified. In the waters off the South Orkney Islands, we recorded the largest number of zooplankton taxa and copepod species; mesozooplankton communities showed its highest diversity ($H' = 2.253$). The mesozooplankton communities in the Bransfield Strait showed its lowest taxonomic diversity ($H' = 0.684$). The total abundance of mesozooplankton ranged from 47.3 to 2840.6 ind./m$^3$, with an average value of $1001.2 \pm 469.4$ ind./m$^3$. The total biomass (wet weight in mg/m$^3$) of mesozooplankton ranged from 20.1 to 13,343.1 mg WW/m$^3$, with an average value of $2643.7 \pm 935.1$ mg WW/m$^3$. The maximum abundance and biomass values were recorded off the South Orkney Islands; the minimum abundance and biomass values were observed in the Bransfield Strait and off the Antarctic Peninsula (Figure 8).

Among the mesozooplankton, copepods (49% of the total abundance and 18% of the total biomass of mesozooplankton) and krill larvae (44% and 68%, respectively) were the dominant mesozooplankton components at all stations. Copepods (*Calanoides acutus*, *Calanus propinquus*, *Metridia gerlachei*, *Oithona* spp., and *Rhincalanus gigas*) were the most abundant in the Bransfield Strait near the South Shetland Islands (74–96% of the total abundance and 88–95% of the total biomass of mesozooplankton), followed by waters off the South Orkney Islands and the Powell Basin (51–56% and 56–60%, respectively). In contrast, euphausiids (krill eggs, nauplii, calyptopis, and furcilia) were the most abundant off the South Orkney Islands and in the Antarctic Sound (79–85% of the total abundance and 75–97% of the total biomass of mesozooplankton), followed by the southwestern and northeastern slopes of the Powell Basin (59–65% and 50–66%, respectively) (Figure 8).

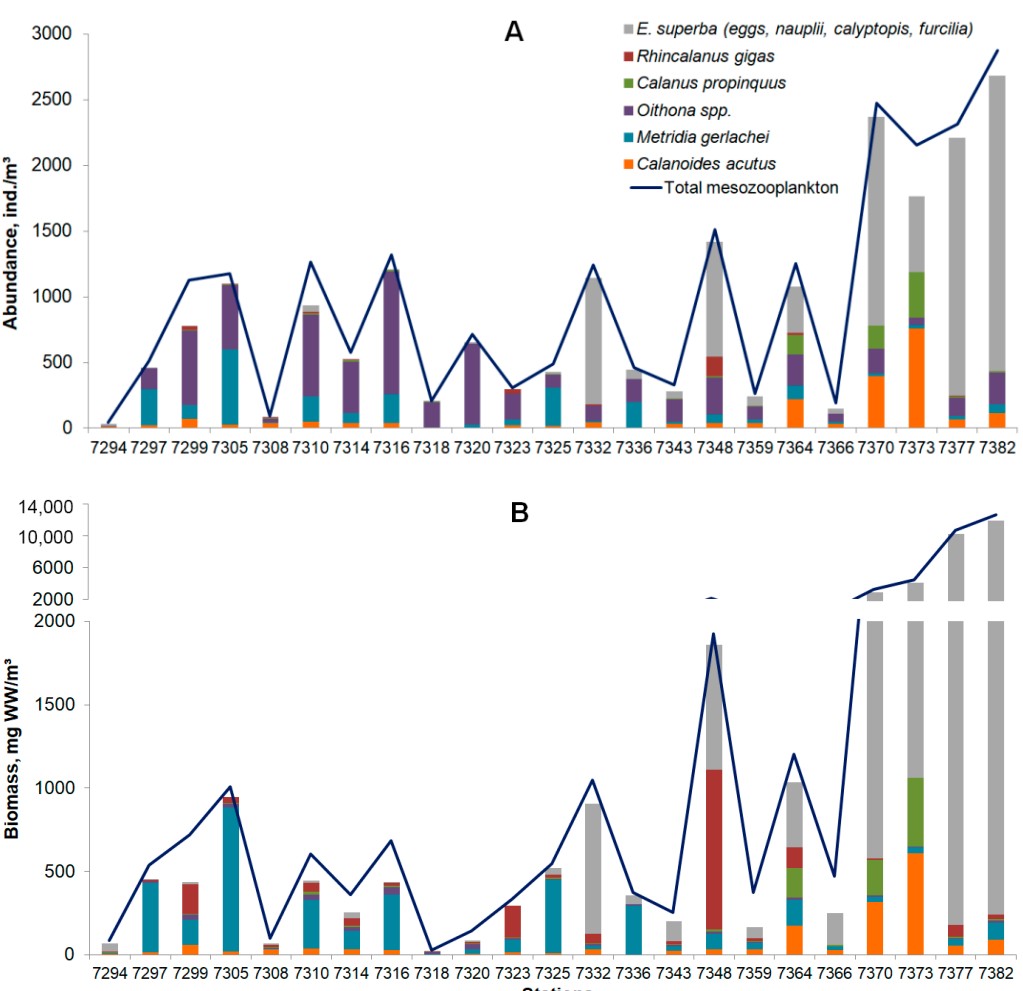

**Figure 8.** Total abundance ((**A**), ind./m$^3$), biomass ((**B**), mg WW/m$^3$), and species composition of mesozooplankton in the study area.

A cluster analysis of the mesozooplankton community revealed a significant similarity between the stations. Four major groups can be distinguished ($R = 0.78$, $p = 0.001$) (Figure 9A). Group A was found in the northern Antarctic Sound and over the southwestern and northeastern slopes of the Powell Basin (Figure 9B); mesozooplankton communities were characterized by copepods *Oithona* spp., *M. gerlachei*, and *C. acutus* (species ranked in the order of descending abundance). Group B was detected at the stations in the Bransfield Strait and off the South Shetland Islands with equal significance to the copepods *Oithona* spp. and *M. gerlachei*. The communities found at the stations between the Antarctic Peninsula and off the South Orkney coast belonged to group C, characterized by krill larvae at different stages of development, followed by copepods *C. acutus*, *Oithona* spp., and *Calanus propinquus*. Group D was represented at stations of the Bransfield Strait and off the Antarctic Peninsula (Figure 9B) and was mainly composed of the copepods *C. acutus* and *Oithona* spp. The stations 7318 and 7336 are exposed to the cold and freshening SW and stand out from the other stations. The communities there were characterized by Antarctic species [77,78] with a relatively low abundance and species diversity.

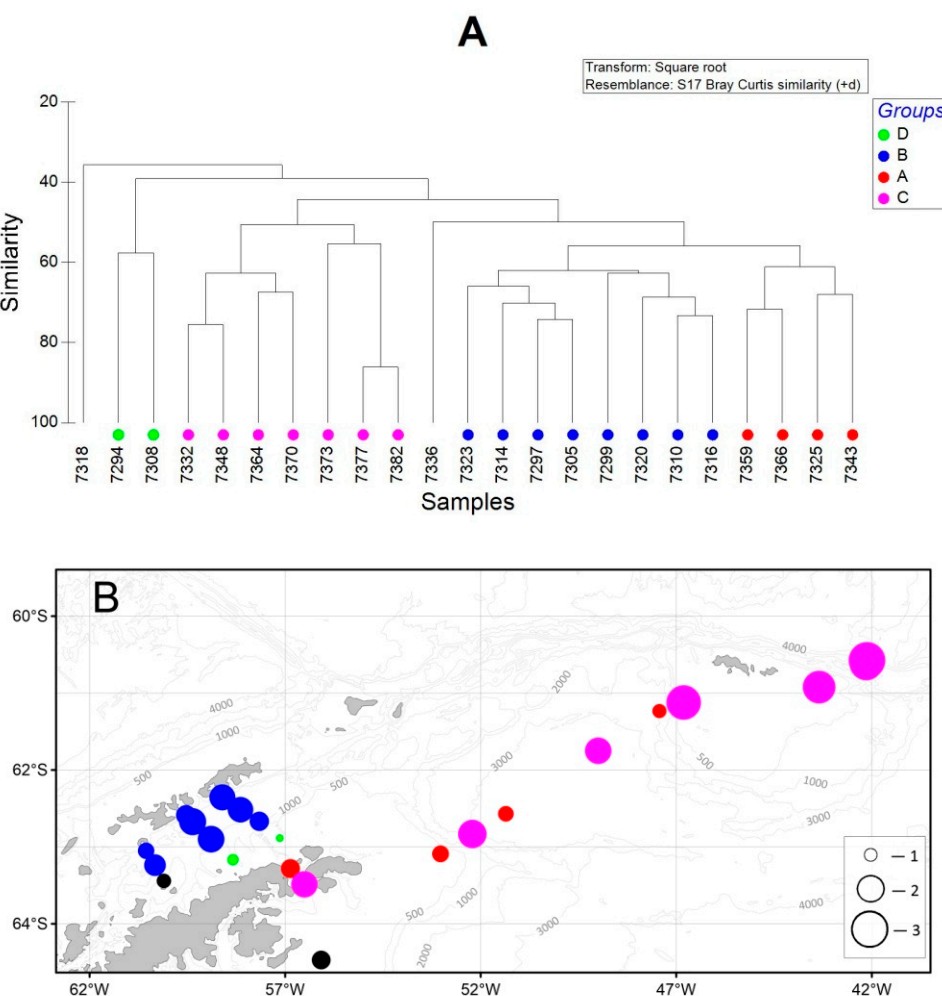

**Figure 9.** Dendrogram of stations resulting from the cluster analysis (**A**) based on the abundance of mesozooplankton and the spatial distribution of stations grouped by cluster analysis (**B**). Groupings: red circle—A, blue—B, pink—C, green—D, black—no grouping. Total abundance (ind./m$^3$): 1—100; 2—1200; 3—3000. Diameter of the circle corresponds to the total abundance at a particular station.

Comparing the locations of the identified communities (A–D) with the hydrological characteristics of the waters, we found that the groups A and C presented in the Antarctic Sound and in the area between the Antarctic Peninsula and the South Orkney Islands were influenced by the modified waters of the Weddell Sea. Group A was influenced by the cold and freshening modified AASW. On the contrary, the communities of the C group were found in deep waters influenced by the warm and salty modified AASW. Group B, occurring in the Bransfield Strait and off the South Shetland Islands, was influenced by the warm TBW and modified mCDW. Group D, occurring in the Bransfield Strait and off the Antarctic Peninsula, was influenced by the cold modified TWW.

Pelagic polychaetes were represented by 16 holopelagic species (plus two morphotypes with an unclear taxonomic level), the epitokous stage of one species, and the larvae of eight benthic species. Their abundance varied from 0.02 to 6.7 ind./m$^3$, 0.7 ± 1.6 ind./m$^3$ on average. The most abundant and widespread species was *Pelagobia longicirrata*; an occurrence > 50% was observed in *Rhynchonereella bongraini*, *Tomopteris arpenter*, *Typhloscolex* sp.1, *Maupasia coeca*, and *Vanadis antarctica*, with the first three of these species along with *P. longicirrata* being the most numerous.

There were no significant differences in species composition between the areas. In terms of the quantitative structure, the southernmost stn. 7336, which became ice-free just prior to sampling (ice margin), and the South Orkney shelf, where an early ice-break was noted, differed significantly (ANOSIM $R = 0.703$, $p = 0.001$) from the main poorly differentiated group of stations (Figure 10A). At stn. 7336, only omnivores *P. longicirrata* and the larvae of one benthic species were recorded (0.35 ind./m$^3$). A well-developed taxocene was found at the S. Orkney shelf. It was characterized by a predominance of mature *P. longicirrata* and *R. bongraini* and the greatest development of carnivores (especially those belonging to the Typhloscolecidae family). The population density was higher than at other stations by one–two orders of magnitude (1.7–6.7 ind./m$^3$, up to 23 ind./m$^3$ in the near-bottom layer). The intragroup similarity was about two–three-times higher than the intergroup similarity (62% to 20–36%, respectively).

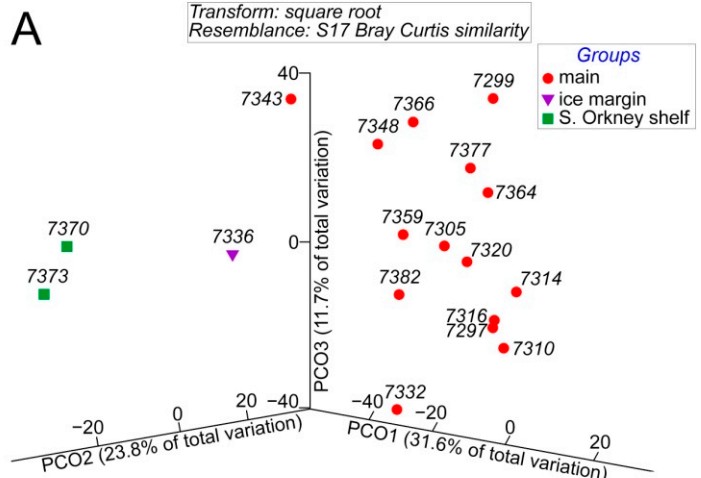

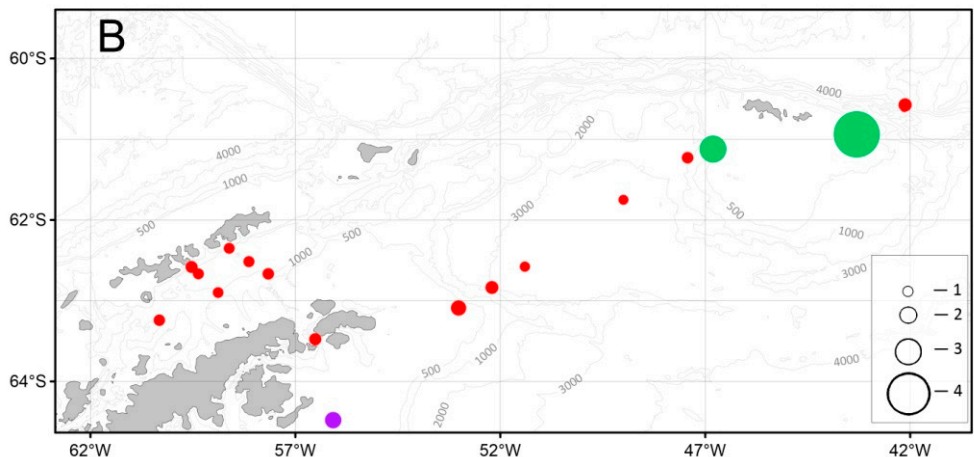

**Figure 10.** Diagram (3D representation) of the PCO ordination of stations by abundance (ind./1000 m$^3$) (**A**) and spatial distribution of stations grouped by the PCO method (**B**). Groupings: red circle—main, violet—ice margin, green—S. Orkney shelf. Total abundance (ind./1000 m$^3$): 1—50; 2—350; 3—1500; 4—6500. See Figure 9 for total abundance designation.

The abundances in the Bransfield Strait (0.07 ± 0.01 ind./m$^3$), at the Weddell Sea shelf (0.19 ± 0.12 ind./m$^3$), and in deep-water areas (0.08 ± 0.05 ind./m$^3$) were low. The number of species was 7 ± 3 per station, except for the stations at the continental slope (up to 13). The intergroup similarity between the areas ranged from negligible to twice as low as the intragroup similarity. Abundance was higher over the shallow depths and minimal over the deeper ones within each major water area (Bransfield Strait, Weddell Sea) (Figure 10B).

The DistLM analysis confirmed that the factors significantly ($p = 0.001$) influencing the quantitative structure of the polychaete taxocene were the time of release of the waters from closed ice and the depth.

### 3.4. Macrozooplankton

Five species of euphausiids were recorded in the study area—*E. superba, E. crystallorophias, E. triacantha, E. frigida*, and *Thysanoessa macrura.* Antarctic krill (41.2% of the total abundance and 46.7% of the total biomass of macrozooplankton) and *T. macrura* (21.7 and 1.9%, respectively) were dominant euphausiids. The tunicate *S. thompsoni* was the most abundant salpa species and accounted for 37.0 and 51.3%, respectively.

The abundance of macrozooplankton varied from 0.1 to 356.6 ind./1000 m$^3$. The maximum abundance of krill (356.6 ind./1000 m$^3$) was recorded at the boundary of packed ice in the zone of the local Chl *a* maximum (station 7336). The maximum abundance of *T. macrura* (52.7 ind./1000 m$^3$) was registered in the eastern part of the Bransfield Strait. The maximum abundance of *S. thompsoni* (201.5 ind./1000 m$^3$) was recorded in the deep waters of the Powell Basin (Figure 11).

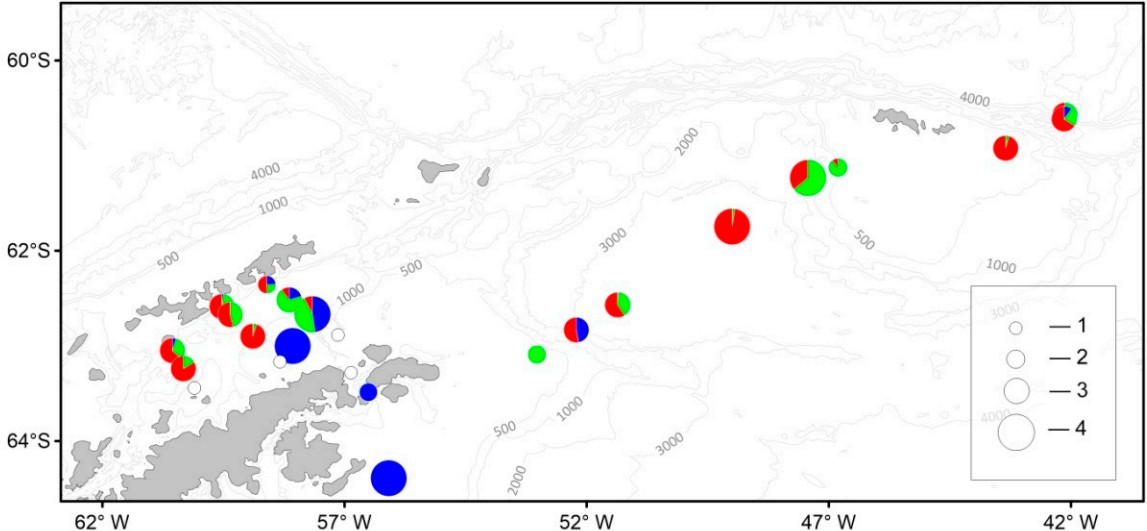

**Figure 11.** Spatial distribution of the macrozooplankton abundance (ind./1000 m$^3$) and contribution (%) of the most abundant species: blue circle—*E. superba*, green—*T. macrura*, red—*S. thompsoni*. Total abundance (ind./1000 m$^3$): 1—0; 2—10; 3—100; 4—400. See Figure 9 for total abundance designation.

The macrozooplankton biomass varied from 0.2 to 207.5 g/1000 m$^3$. In the communities of macrozooplankton in the Bransfield Strait, off the South Shetland Islands, in the Powell Basin, and off the South Orkney Islands, *S. thompsoni* was the predominant species with a maximum relative biomass of 54.8 g/1000 m$^3$ in the Bransfield Strait. The maximum biomass of krill (207.5 g/1000 m$^3$) was recorded south of the Antarctic Sound. The maximum biomass of the *T. macrura* was recorded over the northeastern slope of the Powell Basin (2.8 g/1000 m$^3$) (Figure 12).

Within the surveyed area, two significant groups of macrozooplankton communities A and B were identified with subgroups A1, A2, B1, and B2 (Figure 13A).

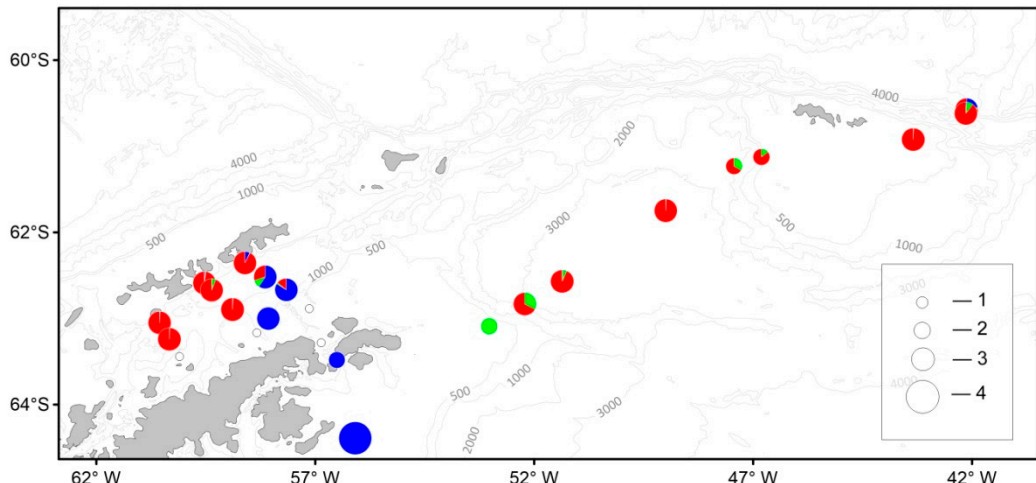

**Figure 12.** Spatial distribution of the macrozooplankton biomass (g/1000 m$^3$) and contribution (%) of the most abundant species: blue circle—*E. superba*, green—*T. macrura*, red—*S. thompsoni*. Total biomass (g/1000 m$^3$): 1—0; 2—10; 3—100; 4—210. See Figure 9 for total abundance designation.

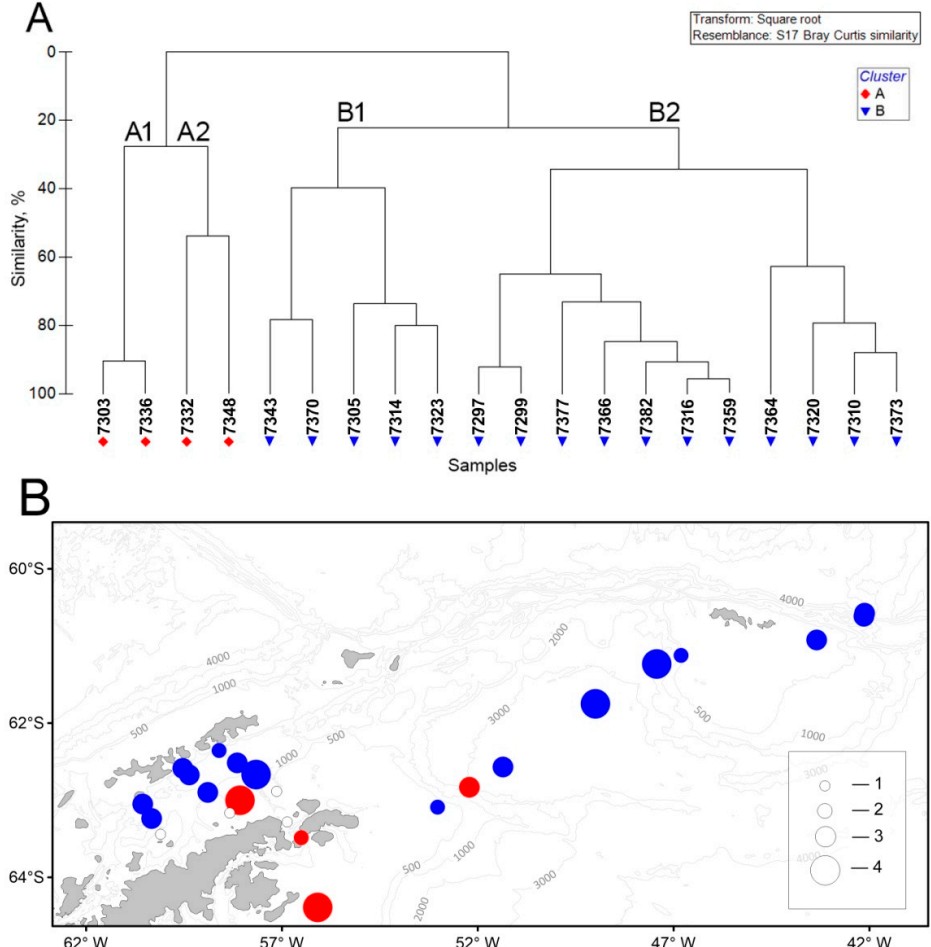

**Figure 13.** Dendrogram (**A**) of the stations resulting from cluster analysis based on the abundance of macrozooplankton and the spatial distribution of stations grouped by cluster analysis (**B**). Groupings: red circle—A, blue—B. Total abundance (ind./1000 m$^3$): 1—0; 2—10; 3—100; 4—400. See Figure 9 for total abundance designation.

Subgroup A1 was located in the southern Bransfield Strait (the shelf of the Antarctic Peninsula tip), as well as in the Weddell Sea (east of the Antarctic Peninsula) (Figure 13B). A feature of this subgroup was a high density of Antarctic krill and the absence of any other macrozooplankton species in the catch. The exception was the station in the Bransfield Strait, where a single specimen of *E. crystallorophias* was found. Subgroup A2 was located in the Antarctic Sound and the Powell Basin. A distinctive feature of this subgroup, in addition to the high density of Antarctic krill, was a significant amount of *S. thompsoni*.

Subgroup B1 was aggregated in the southwestern Powell Basin and in the Bransfield Strait (shelf of the South Shetland Islands) (Figure 13B) and characterized by the presence of a large number of *T. macrura*. Antarctic krill was found in small quantities in catches or was completely absent. *S. thompsoni* dominated at some stations while *E. triacantha* was recorded in the Bransfield Strait. Subgroup B2 included the stations of the Bransfield Strait, the Powell Basin, and the South Orkney Islands. All three dominant macrozooplankton species were present in this subgroup along with *E. frigida*, which was found off the South Orkney Islands.

### 3.5. Ichthyoplankton

Among the ichthyoplankton, 16 species of fish in the early stages of development belonging to eight families (a total of 411 specimens) were found. Larvae and juveniles of the following four species dominated the catches by abundance: Antarctic jonasfish *Notolepis coatsorum*, Antarctic silverfish *Pleuragramma antarcticum*, Antarctic lanternfish *Electrona antarctica*, and Painted notie *Nototheniops larseni*, which together accounted for 77.6% of the total number. The maximum concentrations of ichthyoplankton were observed on the shelf near the eastern tip of the Antarctic Peninsula and northeastward of the South Orkney plateau. On the shelf of the Antarctic Peninsula, the ichthyoplankton was dominated by neritic species of the Notothenidae families (*P. antarcticum*, *N. larseni*, and crowned rockcod *Trematomus scotti*), the total relative abundance of which reached 7.7 ind./m$^2$. Northeastward of the South Orkney plateau, in the Antarctic Circumpolar Current (ACC) zone of influence, the larvae of the mesopelagic species *E. antarctica* and Antarctic deep-sea smelt *Bathylagus antarcticus* were the most abundant, and the total relative abundance of ichthyoplanton in this area reached 5.4 ind./m$^2$. In the Powell Basin, it was noticeably lower and ranged within 0.2–2.5 ind./m$^2$ with the predominance of mesopelagic *B. antharcticus* and *N. coatsorum* in the ichthyoplankton samples. At the stations of the northern and central parts of the Bransfield Strait, under the influence of the TBW, the relative abundance of ichthyoplanton, represented by single specimens of *E. antarctica* and *N. larseni*, was minimal (0.1–0.6 ind./m$^2$).

These features of the distribution of larvae and juveniles of Antarctic fish species are well reflected in the cluster analysis results (Figure 14A). Two (A–B) groups of stations were identified in the studied area. The communities of group A were dominated by mesopelagic species (*E. antarctica*, *B. antarcticus*, and *N. coatsorum*). On the shelf of the Antarctic Peninsula (group B), the larvae of two species of notoneniids were the most numerous, namely, *P. antarctica* and *N. larseni* (Figure 14B).

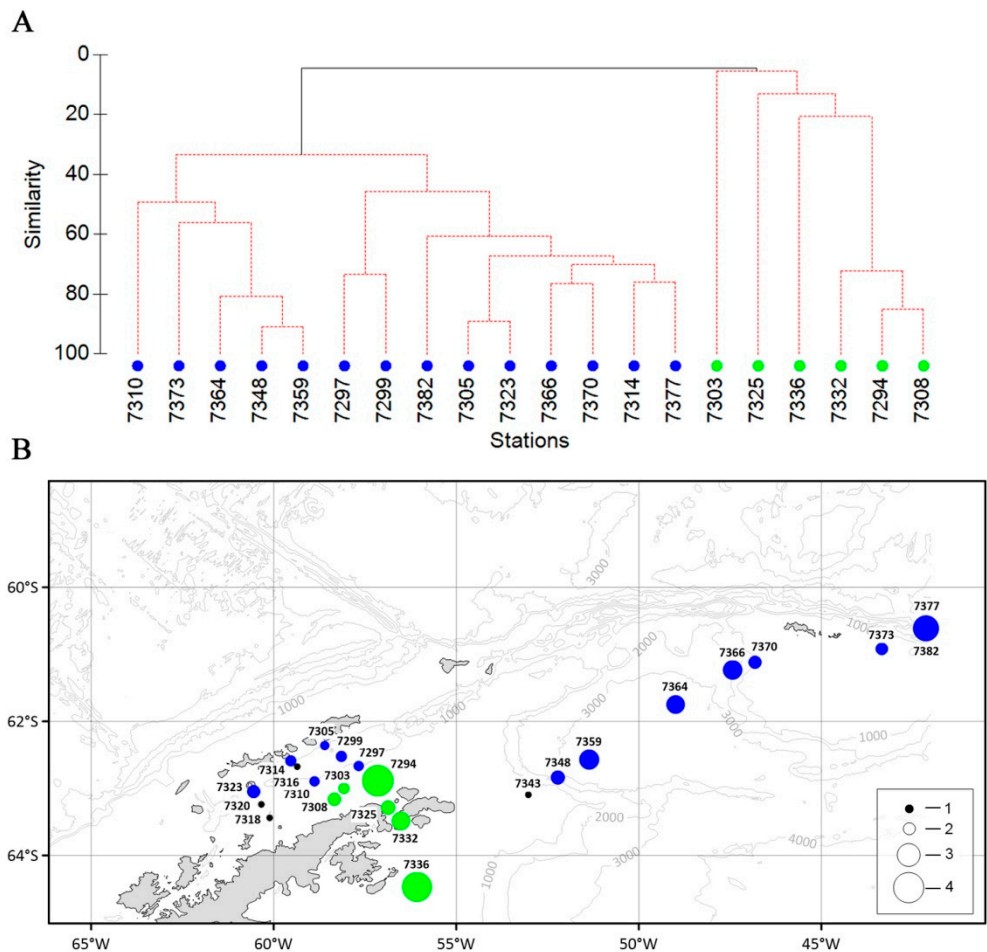

**Figure 14.** Dendrogram of stations resulting from cluster analysis (**A**), based on the abundance of ichthyoplankton and on the geographical distributions of stations grouped by cluster analysis (**B**). Groupings: blue circle—A, green—B. Total abundance (ind./m$^2$): 1—0; 2—1; 3—5; 4—10. See Figure 9 for total abundance designation.

## 4. Discussion and Conclusions

In general, the thermohaline water structure around the Antarctic Peninsula was close to that of recent years [79–83]. The structure of the main currents was also consistent with previous results [39,84,85]. However, some local changes were observed. The first difference was the warmer TWW, in which the potential temperatures below −1 °C were not observed, unlike in recent observations [86]. The Antarctic Sound warmed significantly compared to 2020 [87], while the potential temperature of the other waters around the Antarctic Peninsula did not change compared to the data published by [88]. This indicates the effect of some local processes in the Strait possibly related to the melting ice. A constant flow of water from the Weddell Sea to the Bransfield Strait through the Antarctic Sound was not observed by us; moreover, the structure of the currents was ambiguous, which has also been confirmed by recent studies [87,89]. In the Powell Basin and off the S. Orkney Islands, most of the stations corresponded to the classical distribution of thermohaline properties [80].

Against the background of observed climatic changes, general structural rearrangements of phytoplankton communities were noted. The sets of the characteristic species of the majority of cenoses identified in the austral summer of 2022 consisted mainly of colonial diatom species. In half of the coenotic groups, one of the leading positions belonged to the large colonial diatom *O. weissflogii*. *Chaetoceros* spores played an important structural role for two types of cenoses («Bc» and «Cc»). However, in the summer of 2020 [90], the development of coenotic groups with the predominance of *C. pennatum* and *Thalassiothrix*

*antarctica*, which do not build colonies, was noted. Only in a few areas of the Antarctic Sound were *T. tumida*, *C. gelidus*, and *C. socialis* significant during that season. In the summer of 2022, colonies of *P. antarctica* actively developed in the northern and central parts of the Bransfield Strait and in the southwestern Powell Basin, while, in 2020, mainly its early epiphytic stage was noted [90]. *Porosira glacialis* was not included in any set of the characteristic species of any cenoses in 2020, while, in 2022, it was included in three out of six. In 2020 [90], in contrast to 2022, in some areas of the Powell Basin, a massive development of specific pelagic aggregates formed by profusely reproducing attached heterotrophic flagellates (Kinetoplastida), which used accumulations of living cells and dead valves of the diatom *T. antarctica* as their skeletal basis, was observed.

The distribution of phytoplankton communities in 2022 belonging to different floristic and, in particular, coenotic groups, demonstrated a significant heterogeneity of the composition and structure of the phytopelagial. In some areas, a relatively compact distribution of certain types of communities was observed, which indicated their preferential development in these areas. In the Bransfield Strait, the distribution of the communities of the different groups had a mosaic pattern and, nevertheless, reflected various stages and forms of the metamorphosis of cenoses. Structurally, all these communities, confined to local areas, are variations in the development and multi-vector complexity building of the «Fc» group, which is the simplest in structure and poor in species.

The structure of the studied mesozooplankton communities is typical of the Atlantic sector of the Antarctic [77,78,91]. In general, it has not changed much compared to what is reported in previous studies [17,85,92]. Changes in the structure and abundance of copepods and other mesozooplankton groups in the Southern Ocean are mostly associated with variations in phytoplankton biomass [93]. According to Tarling et al. [94], copepods show a preference for dinoflagellates, confirming that changes in prey quality and food selectivity may be the major factors influencing the stability of mesozooplankton communities.

Previously, Antarctic krill larvae were reported to be transported not only by currents but also by the drift of ice fields, which provides them with food (ice algae) and protection from predators [95,96]. Positive temperatures, the abundance of phytoplankton [97,98], and the presence of ice cover [78,99] are the key factors providing a high density of Antarctic krill larvae. Higher temperatures contribute to an increase in phytoplankton abundance and an earlier arrival of mature krill for spawning [100,101]. Its replenishment from the Weddell Sea and distribution in the Atlantic sector of the Antarctic may be associated with the spawning season of adults [102]. In case of late spawning (February and March), krill larvae cannot survive harsh winter conditions and die, which may result in their low replenishment in the following year [103]. Krill eggs and larvae, which can be carried by the western branch of the Weddell Gyre and mixed with the organisms carried by the waters of the southern ACC jet [104], are regularly found at the boundary between the Weddell and Scotia Seas, off the South Orkney Islands [105]. This may be one of the possible hypotheses for the distribution pattern of Antarctic krill larvae in the study area.

The zoogeographic boundaries of pelagic polychaetes in the West Antarctic are associated with the Antarctic Convergence and the boundary of the Bellingshausen and Weddell Seas [106,107], which are beyond the scope of our study area. Lana and Blankensteyn [108] expressed a more radical opinion: pelagic polychaetes are not indicators of water masses in the West Antarctic, and their distribution depends mainly on seasonal or interannual dynamics. This is confirmed by the fact that various scientists [107,109–111]; this study have reported a similar set of species; however, their spatial distribution differs greatly between the seasons. Our data confirm that a common pool of species was observed in the study area. The relationship between the quantitative and functional structures of the taxocene and the time of the release of the studied waters from the closed ice showed that the differences between the stations are due to seasonal succession. The increase in the importance of the species of the Typhloscolecidae family from south to north is particularly expressive. These organisms require a successive development of herbivorous

zooplankton along with chaetognaths, consuming it, which are, in turn, the obligate prey of typhloscolecids [112,113].

Due to the complex hydrological structure of the Bransfield Strait, it would be logical to expect a complex structure of the taxocene of pelagic polychaetes, corresponding to the distribution of water masses. However, the uniformly low abundance along with a developed functional structure, which was also observed by other authors [108,111], is apparently due to a less pronounced seasonal succession associated with the absence of winter ice cover. An increase in species richness over the continental slope is explained by the presence of a frontal zone associated with slope currents in such areas [114], which forms a more diverse biotope.

The abundance of mass species *P. longicirrata* and *R. bongraini*, whose diet is dominated by phytoplankton [115,116] and negatively correlated with the concentration of Chl *a* and, consequently, with the abundance of most phytoplankton functional groups. This was due to the differences in the predominant group of microalgae across the areas. Large forms armed with spikes or setae, as well as extensive colonies of *P. antarctica*, which prevailed at the southern stations of the Weddell Sea rich in phytoplankton, are practically not recorded in the stomachs of *P. longicirrata* and *R. bongraini* [117–119]. At the South Orkney shelf, small-celled diatoms of a more compact form, which usually predominate in their diet, were abundant. This determined the maximum abundance of omnivorous/herbivorous polychaetes in this area. The trend towards an increase in the role of small-celled forms in the Antarctic phytoplankton, noted by several authors [120–122] and this study, should create favorable conditions for these species in the future.

The current climate changes result in significant fluctuations in atmospheric phenomena and hydrophysical conditions that affect the composition and structure of water masses, sedimentation processes, and the duration of the ice-free period. This leads to cascading structural and functional changes in the Antarctic marine ecosystem, affecting the duration of the phytoplankton bloom period, the quantity and distribution and ratio of biogenic elements, and impacts on the structure and productivity of the Antarctic ecosystem as a whole [17–19,123]. Yang et al. [124] carried out a circumpolar analysis of the data on the dynamics of the Antarctic krill abundance and distribution and concluded that a notable decrease in its abundance was characteristic of the Atlantic sector of the Southern Ocean, while the density in other parts of the range did not change over time. In our study area, the average abundance and biomass of Antarctic krill, according to our data, turned out to be two- to three-times lower compared to the data of other researchers [125]. However, the main areas of its accumulations remained the same (the inshore shelf waters off the Antarctic Peninsula) within the zone of the ACoC influence, carrying cold and salty TWW from the Weddell Sea.

Our data on the maximum abundance and dominance of Antarctic krill among other species of macrozooplankton in the Bransfield Strait (the Antarctic Peninsula shelf), as well as to the south of the Antarctic Sound, is associated with a combination of the most favorable environmental conditions, namely the water temperature, Chl *a* concentration, and salinity. Thus, in the Bransfield Strait, these conditions are formed due to the presence of TWW, where the temperatures range from $-0.8$ to $-1.7\,°C$, which is within the optimal temperatures for the growth and development of both juveniles and mature Antarctic krill [126–130]. Earlier, based on the identification of several groups of juvenile Antarctic krill, it was suggested that there are waves of "replenishment" of its juveniles locally in the Bransfield Strait area transported by the western branch of the Weddell Gyre [96]. The data collected by us indicate the transportation of Antarctic krill from the Weddell Sea to the Bransfield Strait with a cold ACoC. Its main biomass in the Antarctic Sound is also formed by organisms brought from the Weddell Sea. The modeling (nMDS analysis) of the relations between the optimal basic environmental conditions (water temperature, salinity, Chl *a*, oxygen content) and the spatial distribution of Antarctic krill [131] revealed that, for all stages of its development, the optimal temperature for different depths was $0.1–0.4\,°C$.

At the same time, the model indicates that immature individuals inhabit slightly colder waters compared to mature ones.

It is known that juvenile Antarctic krill is more stenothermic and prefers a temperature range of $-1.0$–$1.5\,^\circ$C compared to mature individuals that can withstand a wider temperature range of up to $4\,^\circ$C [132,133]. We compared the catches from the Bransfield Strait, the Antarctic Sound, and the area to the east of the Antarctic Peninsula (in descending order) with other surveyed areas. As a result, a significant number of juveniles of different sizes were found. At the same time, at Stn. 7336, the abundance of Antarctic krill indicated maximum values with minimal salinity (33.55 psu), low temperatures ($0.1\,^\circ$C), and high Chl *a* concentration ($>20\,$mg/m$^3$) in the surface layer; while, at a depth of 200 m, the potential temperature was close to the freezing temperature of sea water ($-1.8\,^\circ$C), salinity was higher, and the Chl *a* concentration did not exceed $0.5\,$mg/m$^3$. As in our studies, Nishikawa et al. [134] noted a high abundance of Antarctic krill in areas with high Chl *a* concentrations.

Changes in the abundance and responses of the population of *S. thompsoni* to environmental changes are still insufficiently studied [135–137]. Our data indicate a large current outbreak of salps in the study area. A sharp increase in their abundance in the Bransfield Strait was promoted not only by the warm BC, but also by their feeding habits, energy saving reactive movement, and a complex reproductive cycle [138,139]. In the Powell Basin, as well as off the South Orkney Islands, a favorable water temperature regime (with warming up to $2\,^\circ$C) for the development of salps was recorded. Unlike Antarctic krill, the development of salps does not depend on the ice cover and associated algae, which are the main prey of krill larvae and juveniles [30]. At the same time, we recorded a high density of *T. macrura*, a species which inhabits similar hydrophysical conditions and competes with salps for forage [123,140]. Unlike Antarctic krill, whose lifespan can be over 5 years, *S. thompsoni* lifespan varies from 3 months to 2 years [123], and fluctuations in their abundance reflect the annual variability of the environmental conditions that contribute to the explosive rise in their abundance [21,26]. For instance, several studies have noted outbreaks of salps after winters with a relatively low sea ice development [14].

In previous decades, many works examined the species composition and distribution of ichthyoplankton in the Antarctic Peninsula area [141–148] and in the northern part of the Weddell Sea in the Powell Basin [149–151]. The presence of both pronounced seasonal and interannual [143,144] and spatial variability [152] in the composition and the abundance of Antarctic fish communities at the early stages of development were observed. In total, more than 40 species were found in the ichthoplankton of the examined area [146,153]. In the coastal waters and on the shelf of the Antarctic Peninsula, the Notothenidae and Channichthyidae families dominated by species numbers and total abundance, among which *N. larseni*, dusky rockcod *Trematomus newnesi*, *P. antarcticum*, and ocellated icefish *Chionodraco rastrospinosus* occurred with maximum frequency [142,143,146]. Outside of the neritic zone, they were replaced by mesopelagic species, mainly from the Myctophidae, Paralepididae, and Bathylagidae families [150,151]. As a rule, there was no sharp boundary between communities, and the representatives of both ecological groups were often present in the same samples. The species composition and distribution pattern of ichthyoplankton in our collections are largely similar to that of some previous studies in this area in the austral summer [142,149–151]. In terms of differences, firstly, massive concentrations of nototheniids occurred exclusively to the east of the tip of the Antarctic Peninsula in areas affected by the waters of the Weddell Sea, and, secondly, they were completely absent on the shelf of the South Shetland Islands and in the northern and central parts of the Bransfield Strait, where the water masses of the Belingshausen Sea prevailed. This distribution pattern is not typical for this area. As a rule, the larvae and juveniles of nototheniids are represented throughout the entire waters of the Bransfield Strait, although, in some years, a similar pattern of their distribution took place [144,148]. The observed differences can be explained both by the use of different fishing gear and by the uneven distribution of the objects of study, as well as by the variability of oceanographic conditions. Unlike in the Bransfield Strait, the structure of the ichthyoplankton community in the Powell Basin

fully corresponded to the average long-term situation with the dominance of mesopelagic *E. antarctica*, *N. coatsorum*, and *B. antarcticus.*

As a result, the study of the thermohaline structure of the upper 300-m water layer in the examined area demonstrated, as before, no significant deviations in the water structure. The main difference from some previous observations is a relative warming of the TWW and its stronger southward propagation than previously thought. The waters of the Antarctic Sound have become warmer, and a constant flow of water from the Weddell Sea to the Bransfield Strait through it has not been recorded. In the other waters around the Antarctic Peninsula and in the Powell Basin, no important changes in the thermohaline structure have been revealed. Changes in the Bransfield Strait and Antarctic Sound may have a significant impact on the entire region and require further observations.

There have been significant structural changes in phytoplankton communities in the summer of 2022 compared to 2020, which manifested mostly in their complexity. A significant role in the spatial organization of the cenoses of different types was played by the colonies of *O. weissflogii*, *C. gelidus*, and *C. socialis* (in many areas at the stage of mass sporulation), and *C. tortissimus*, *T. rotula*, *P. glacialis*, *F. sublinearis*, *F. kerguelensis*, *P. antarctica*, and non-colony forming *C. pennatum*. In the 2020 season, pelagic microaggregates played a significant role in the structure of the phytopelagial in many areas as loci for the joint development of bacterial colonies and benthic and planktonic microphyte flora [90], while, in the summer of 2022, colonial microphytes outside the aggregates dominated in many areas. The structuring role in the formation of aggregates in 2022 often belonged to the extensive free-floating colonies of *P. antarctica.*

Against the background of a significant reorganization of the structure of phytoplankton communities, there have been significant changes in the size and morphological structure of the Antarctic krill forage resources. In fact, they lost some mass diatoms (*O. weissflogii*, *C. pennatum*, *Proboscia* spp., *Rhizosolenia* spp., *F. sublinearis*, large-celled form *P. glacialis*), which were not consumed since krill was either unable to feed on them or could only do so in small quantities due to their large size [154–156]. Krill, as well as salps, is also unable to feed on pelagic aggregates based on large colonies of *Phaeocystis* [157,158]. Nevertheless, against this background, salps that consume a wider range of various life forms of phytoplankton could gain a competitive advantage [158]. At the same time, it is known that salps cannot effectively digest most of the captured diatoms due to the lack of morphological structures necessary to break their silica shells [159,160].

Despite the warming of the surface layer of water in the study region, the major species of copepods and other groups of mesozooplankton have generally retained the pattern of their spatial distribution during the austral summer. The detection of two different mesozooplankton communities in the Bransfield Strait confirms the existence of the known multidirectional two-jet system in this area, consisting of a powerful flow of warm water from the BC and a cold water from the ACoC. Small copepods *Oithona similis*, as well as large copepods of the genera *Calanoides*, *Calanus*, *Rhincalanus*, and *Metridia*, dominated in terms of abundance, while Antarctic krill larvae at different developmental stages dominated by biomass. Currently, mesozooplankton communities of the region inhabit the waters with a higher surface temperature than many decades ago. However, no shifts in the ranges of the dominant taxa have been recorded to date.

There are no distinct spatial groups or taxocenes of pelagic polychaetes in the study area. The most widespread species are omnivores with a predominance of phytoplankton in their diet. However, their distribution was negatively correlated with the abundance of phytoplankton, as it was instead determined by the composition of the predominant microalgae species. Carnivorous species' abundance, on the contrary, followed the trend in abundance of their prey, showing no relation with its species composition. This is expectable considering that the observed carnivorous polychaete species consume any prey with a suitable body size. As for the most specialized group, Typhloscolecidae, no preference for certain species of chaetognaths has been shown so far, and a high abundance of the latter group was typical for all stations, with the later succession stage showing

higher abundances of typhloscolecids. Thus, the observed differences between stations followed the overall seasonal succession, the best proxy for which is the time of the waters' freeing from closed ice.

Our results demonstrate that the studied macrozooplankton species are tied into a certain combination of hydrophysical factors, among which the major abiotic factor was the surface temperature and the major biotic one was the availability of forage resources. The circulation of water masses in the region was also important in these dynamics. With regards to the optimal conditions for the maximum concentrations of Antarctic krill, they only occur in the Bransfield Strait and south of the Antarctic Sound, on the boundary of packed ice and in a relatively warm and freshened surface layer, where a combination of cold and salty subsurface waters exist. In addition, relatively low krill replenishment in the study area may be associated with a late krill spawning period in 2021; therefore, the larvae could not survive the winter. Data on the abundance of Antarctic krill in the last decade indicate its significant fluctuations in the Atlantic sector of the Antarctic [23,154]. Previous studies reported the stability of Antarctic krill stocks [161]. Such discrepancies may be associated with different approaches to measuring the abundance and biomass of Antarctic krill; therefore, a standardization of research methods is required [162,163]. In addition, these contradictions might be due to variations in the lifecycles of the organisms [21,26], as well as due to the mobility and active movements of krill individuals avoiding fishing gear [164,165]. Recent and other observations [166] indicated an increase in the number of salps and an extension of their range, resulting in forage competition with Antarctic krill and other euphausiids and a shift in their ranges poleward towards cold waters.

Among the peculiarities of the distribution of ichthyoplankton in the surveyed area in January–February 2022, the absence of the larvae of nototheniids in the northern part of the Bransfield Strait and their concentrations off the eastern tip of the Antarctic Peninsula can be noted. The observed differences could most likely be explained by the variability of oceanographic conditions or the uneven distribution of the objects of study. Therefore, it can be concluded that our ichthyoplankton survey did not reveal any significant deviations in the species composition and abundance of fish at the early stages of development off the Antarctic Peninsula and in the Powell Basin, which could be explained by modern climatic trends.

Aggregations of phytoplankton (*Actinocyclus*, *Asteromphalus*, and *Fragilariopsis*), meso-zooplankton, pelagic polychaetes, euphausiids *T. macrura*, and tunicate *S. thompsoni* found in the waters of the South Orkney Islands with high values of water temperatures and with low values of salinity were more abundant than these groups in the shelf waters off the Antarctic Peninsula, whereas the distribution of phytoplankton (*Chaetoceros*, *Odontella*, *Phaeocystis*, and *Corethron*), krill, and the larvae of fish are mainly located in the waters off the Antarctic Peninsula with low temperatures, high values of salinity, and a high Chl *a* concentration. Previous studies have reported a substantial increase in Chl *a* occurring farther south of 63° S [167]. Such changes are associated with changes in the composition of the phytoplankton community and are consistent with the expected widespread consequences of marine warming [167,168]. We observed a significant reorganization of the structure of phytoplankton communities and a decrease in the mass of diatoms relative to other phytoplankton. A reduction in diatoms in the diet of Antarctic krill is likely to reduce both growth and reproduction [97,126,127].

The changes we observed in the composition of plankton communities in Antarctic ecosystems (replacement of some dominant groups by others, restructuring, and an increase in the complexity of the structures), as well as widening of the area of the warmer surface waters, can lead to an increase in the abundance and stability of populations of previously less numerous species in the region. These preliminary results of our comprehensive study, which hopefully will be analyzed in detail in subsequent publications, can provide invaluable information about the changing biota of the Southern Ocean. According to climate models [136,169–172], the continued warming of the climate, the retreat of glaciers, and a decrease in the area and thickness of the ice cover can have irreversible consequences

for the succession of plankton in the ecosystems of the Southern Ocean. Any recorded trend relates only to the analyzed time period, and additional evidence is needed to distinguish it from the variability acting on longer time scales [22]. Ultimately, the results of current and future studies will be crucial for monitoring long-term changes and for the modeling of the Antarctic pelagic ecosystem. To forecast possible changes in plankton communities in the future, annual data on the structure, quantitative characteristics, correlation with the hydrophysical conditions, and climate signals, including the interannual variability, which is necessary both for the conservation of the unique biodiversity of fragile Antarctic ecosystems and the rational exploitation of their biological resources by humans, are needed.

**Supplementary Materials:** The following supporting information can be downloaded at: https://www.mdpi.com/article/10.3390/d14110923/s1, Table S1: List of stations conducted onboard the 87th cruise of the R/V Akademik Mstislav Keldysh during January–February 2022 in the Atlantic sectors of the Southern Ocean.

**Author Contributions:** A.M.O. and V.V.K. conceptualized and designed the study; A.M.O., V.V.K., D.G.B., O.A.Z., A.V.M., S.A.M., O.Y.K., V.L.S., G.D.K., V.P.V. and E.S.C.; collected samples and investigation; A.M.O., V.V.K., D.G.B., O.A.Z., A.V.M., S.A.M., P.V.S. and V.L.S.; analyzed the data and wrote the first draft of the manuscript; A.M.O., V.V.K., S.A.M. and V.L.S.; reviewed and edited the MS. All authors have read and agreed to the published version of the manuscript.

**Funding:** This research was funded by State Task FMWE-2022-0001 by the Ministry of Science and Higher Education of the Russian Federation and the State Task for NSCMB FEB RAS no. 122072000067-9, for IBSS RAS no. 121090800137-6, for KarRC RAS FMEN-2022-0006 and by the Russian Science Foundation grant no. 22-77-10004 (CTD data analysis).

**Institutional Review Board Statement:** Not applicable.

**Informed Consent Statement:** Not applicable.

**Data Availability Statement:** Not applicable.

**Acknowledgments:** The authors are grateful to the administration of the Shirshov Institute of Oceanology, Russian Academy of Sciences for the organization of the expedition and field operations, to the plankton and benthos research teams for collecting and providing plankton samples, and also to the captain and the crew of the R/V *Akademik Mstislav Keldysh* for comprehensive assistance. The authors thank anonymous reviewers for their valuable comments and suggestions which allowed for considerable improvement of the manuscript.

**Conflicts of Interest:** The authors declare no conflict of interest.

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
