# Peer review of "Composition and Distribution of Plankton Communities in the Atlantic Sector of the Southern Ocean"

_diversity, doi:10.3390/d14110923_

Round 1

Reviewer 1 Report

see file attached

Author Response

We are deeply thankful for consideration of the manuscript and valuable comments and suggestions made by reviewers that have greatly improved the manuscript. We provide point-by-point answers to comments of reviewers and highlighted changes in the manuscript using tracked changes. We hope that you will find our responses to reviewers’ comments satisfactory, and that the manuscript may now be acceptable for publication.

Response to Reviewer 1 Comments

Point 1: To start with the title does not reflect of the paper content: first, it should be composition and distribution (composition and structure are too close in their meaning); second, “under conditions of continuing climate warming” is hardly discussed in the paper. It is appropriate to talk about it in the discussed (which authors do!) but if it is in the title, it means that data should be provided in the paper to be able to compare past and present oceanographic data sets to reveal or document changes occurred. It is up to authors indeed but I think that the title could be reflective of the content: “Composition and distribution of plankton communities in the Atlantic Sector of the Southern Ocean”.

Response 1: The title “Composition and Structure of Plankton Communities in the Atlantic Sector of the Antarctic under Continuing Climate Warming” was changed to “Composition and Distribution of Plankton Communities in the Atlantic Sector of the Southern Ocean”

Point 2: English is generally good but in some places it requires corrections. Thus, I would recommend spell check the entire text again. For example, it seems that in lines 629-630 some text is missing: “Its re ??? from Weddell Sea and...”

Response 2: The sentences Ln 629-630 was corrected to “Its replenishment from the Weddell Sea and…”

Point 3: Graphs are generally well prepared. However, it is strongly recommended that a figure to be added for macrozooplankton similar to figure 8 presented for mesozooplankton. This would be very good for comparative purposes and would create another line of discussion when both meso- and macrozooplankton densities compared.

Response 3: If it possible, we would like to have figure 8 as it is. The purpose of such presentation of the obtained results is that it will be visually more convenient to use distribution maps, rather than graphs in this case. First of all, this is one of the most common forms of data visualization for euphausiids and some other species in this region. Thus, our data can be compared with others presented in a similar way. Second, we discuss three most abundant species of macrozooplankton (other remaining species were negligible) and their distribution over the area with reference to hydrophysics and partly to phytoplankton. It is also more convenient and correct to compare data presented on maps.

Point 4: It is not clear to what depth mesozooplankton data were used in this publication? It is mentioned that multinet (depth stratified sampling to 1000 depth) and WP-2. Was the latter also used to collect depth stratified sampling? Are the mesozooplankton data presented here for the whole depth sampled or to a particular depth? It should be clearly described. By the way, it is customary and solve lots of problems when a table created with information on stations samples, which gears used, to which depth sampling was carried out and so on. This table can be placed into a supplementary appendix.

Response 4: The sentences Ln 101-104 was corrected to “… Multinet opening/closing net system … by performing vertical tows from 500 m to 300, 300–200, 200–100, 100–50 and 50–0 m. At the non deep-sea stations (maximum depth 200 m), sampling was performed by vertical tows of a WP-2 net of 150 μm mesh-size [36] from 3 m above the bottom to the surface”

We added table into a supplementary materials with information on stations where samples were collected.

Point 5: It is not clear why besides main groups, e.g. copepods, euphausiids, salps, only polychaetes were singled out? How about hyperiids, chaetognaths, pteropods…? If there are target groups to be described in the paper, it should be clearly indicated upfront.

Response 5: Ln 161, We added the sentences “Copepods and krill larvae, as a dominant group of mesozooplankton, were subjected to a more detailed processing. Holopelagic polychaetes, as a less abundant, but rare and significant group of mesozooplankton, were subjected to a more detailed taxonomical processing. Other mesozooplanktonic groups, e.g. hyperiids, chaetognaths, pteropods were found in the study area occasionally and therefore not described, but their abundance was evaluated and added to the total abundance. Euphausiids and salps, as a very abundant group of macrozooplankton, were subjected to a more detailed processing.”

Holopelagic polychaetes are considered a part of mesozooplankton. Their species identification is difficult and they are usually few in samples [Ushakov, 1972; Gagaev and Kosobokova, 2012]; consequently, they are rarely used in describing zooplankton communities [see, for example, Boysen-Ennen and Piatkowski, 1988; Boysen-Ennen et al., 1991; Dauby et al., 2001 and others]. However, they play a significant role the organic matter fluxes [Guglielmo et al., 2014], and may reach high densities, especially in polar regions [Mileikovsky, 1962, 1969; Yingst, 1974; Maurer and Reisch, 1984]. On the other hand, their zoogeographical and ecological peculiarities result in distribution patterns which are often distinct from those of the major zooplankton groups [Tebble, 1960; Lana and Blankensteyn, 1987]. Thus, whereas omitting or coarsening data on this group brings to a considerable underestimation of the overall diversity and abundance of zooplankton; combining them with the major mesozooplankton dataset increases heterogeneity of the data and introduces additional noise. The latter is a usual case when combining together groups differing in basic functional traits [Warwick, 2014].

Point 6: Conclusions should be combined with the discussion.

Response 6: Done

Point 7: The discussion should be refocused on the complexity of physical environment that should be overlaid by biological data. This is largely done in the paper but currently there is very weak link between plankton communities, from phytoplankton to macrozooplnkton. There are different assemblages and often they correlate or not between different plankton groups. There is a good opportunity to make a tighter comparison of the whole plankton assemblage in discussion.

Response 7: We made such discussion and some assumption for macrozooplancton, for example. We discuss abundance and distribution of the studied species in association with certain waters differed in main hydrophysical parameters, including the concentration of Chl a. More, we referred to our first data on the modeling (nMDS analysis) of the relations between the optimal basic environmental conditions (water temperature, salinity, Chl a, oxygen content) and the spatial distribution of Antarctic krill [131] that is characteristic for all stages of its life cycle, e.g. the optimal temperature for different depths was 0.1-0.4°C. At the same time, the model indicates that immature individuals inhabit slightly colder waters compared to mature ones.

Point 8: I would recommend to use more efficiently published sources in the discussion. This part of the Southern Ocean is probably the best studied in the Southern Ocean and there are dozens of publications describing zooplankton communities in the area of investigation. It will become obvious from the literature for example that salps and T. macrura (as well as E. frigida and E. triacantha) belong to the same pelagic assemblage. Also, salps life cycle can range from 3 months to 2 years…

Response 8: Dear Reviewer, please consider the reference list of the manuscript, of which approx. 40 papers we used to discuss macrozooplancton in particular. Among them Johnson et al., 2022 [123] is a huge monograph summarizing comprehensive data on Antarctic ecosystem and dominant species. We used also classical references such as Murphy et al., 2004 [102], Reiss, Ross et al. Contemporary and recent ones as Spiridonov et al. The list of literature used in the manuscript is quite large and we are not in favour to expand it even more. In the discussion part, we tried to use the most up-to-date information. In particular, we used such complex works as Siegel, 2016; Pakhomov, et al., 2020; McBride, et al. 2021; Johnston, et al., 2022. We added few changes in discussion/conclusion part to improve the text.

Reviewer 2 Report

diversity-1944138

Composition and Structure of Plankton Communities in the Atlantic Sector of the Antarctic under Conditions of Continuing Climate Warming

General comments:

The present study presents relevant information about plankton communities in the Atlantic sector of the Antarctic.  The authors aimed at monitoring the phytoplankton, mesozooplankton, microzooplankton and ichthyoplankton under the current climatic changes, focusing specifically on the Antarctic krill that is a valuable commercially exploited species, and also most of the studied area have been designated as Marine Protected Areas. I believe the study is of extreme importance, since these ecosystems should be monitored and studied constantly, as they are a region considered one of the most unstable ecologically fragile of earth, and subject to changes related to climate variability.

In general, the paper is well written and structured. However, I found that there are some changes that should be considered before publication in Diversity. To improve the manuscript, changes and suggestions are described below in the specific comments for all sections of the paper. Also, I would recommend a revision of the English language by a native speaker.

Specific comments:

Abstract

Page 1, Line 32: I suggest changing to: “Euphausiids were found in low abundance, species diversity, and biomass.”

Introduction

Page 2, Line 64: I would write Salps or Salpa, not Salpas.

Page 2, Line 73: Correct the sentence to: “Unlike krill, salps do not depend on ice cover for their development, nor do they need…”

Materials and Methods

I would put the section “Oceanographic measurements” before the biological sampling, the same order as you mention in the results.

Page 4, Line 170: Correct to: “…was carried out down to the species level whenever possible…”

In terms of the data analysis, I would like to know why you used different techniques among some groups. For instance, why didn’t you calculate the diversity index and a factor analysis performed using the distance-based linear models (DistLM) routine for other groups like the macrozooplankton? In fact, you mention in your abstract the low diversity of Euphausiids.

Results

Page 9, Line 324: I suggest changing to: “Results of phytoplankton communities revealed…”

Page 13, Lines 453-462: This paragraph seems more like a discussion, for instance, when you mention terms like “we can conclude”. Is this the result of the factor analysis? You should mention it here.

Related to Figure 10, are these results of all the mesozooplankton community? I was a little bit confused, because when you describe the results in the text referring to this Figure, you only mention the polychaeta species.

Discussion

Page 17, Line 629: It seems like there is something missing in the beginning of this sentence.

Conclusions

Regarding the conclusions, I believe most of what you have written, are more like a discussion. Maybe try to shorten a little bit the conclusions section and use some of the information in the discussion.

Author Response

We are deeply thankful for consideration of the manuscript and valuable comments and suggestions made by reviewers that have greatly improved the manuscript. We provide point-by-point answers to comments of reviewers and highlighted changes in the manuscript using tracked changes. We hope that you will find our responses to reviewers’ comments satisfactory, and that the manuscript may now be acceptable for publication.

Response to Reviewer 2 Comments

Point 1: Page 1, Line 32: I suggest changing to: “Euphausiids were found in low abundance, species diversity, and biomass.”

Response 1: Done

Point 2: Page 2, Line 64: I would write Salps or Salpa, not Salpas.

Response 2: Done

 Point 3: Page 2, Line 73: Correct the sentence to: “Unlike krill, salps do not depend on ice cover for their development, nor do they need…”

Response 3: Done

Point 4: I would put the section “Oceanographic measurements” before the biological sampling, the same order as you mention in the results.

Response 4: The section “Oceanographic measurements” was moved prior to the biological sampling.

Point 5: Page 4, Line 170: Correct to: “…was carried out down to the species level whenever possible …”

Response 5: Done

Point 6: In terms of the data analysis, I would like to know why you used different techniques among some groups. For instance, why didn’t you calculate the diversity index and a factor analysis performed using the distance-based linear models (DistLM) routine for other groups like the macrozooplankton? In fact, you mention in your abstract the low diversity of Euphausiids.

Response 6: Groups of zooplankton animals had specific distribution patterns across the study area which was influenced by different factors for each group. As a result, reliability of methods differed between particular groups. Eventually, we implemented methods that gave most significant results in each case.

We are mainly dealing with Euphausiids – E. superba and T. macrura, only two major species, and tunicate – salp, single species. Other species were negligible unabundant. So, we analyze similarity between stations based on quantitative data (density of animals in a sample) using the Bray-Curtis index. To assess the reliability of clustering, the SIMPROF permutation test (number of permutations 999, p = 0.05) was performed. Same analysis was applied for mesozooplankton as well.

 Point 7: Page 9, Line 324: I suggest changing to: “Results of phytoplankton communities revealed…”

Response 7: Done

Point 8: Page 13, Lines 453-462: This paragraph seems more like a discussion, for instance, when you mention terms like “we can conclude”. Is this the result of the factor analysis? You should mention it here.

Response 8: The sentences “Comparing the locations of the identified communities (A–D) with the hydrological characteristics of the waters, we can conclude…” was corrected to “Comparing the locations of the identified communities (A–D) with the hydrological characteristics of the waters, we found…”.

 Point 9: Related to Figure 10, are these results of all the mesozooplankton community? I was a little bit confused, because when you describe the results in the text referring to this Figure, you only mention the polychaeta species.

Response 9: Holopelagic polychaetes are considered a part of mesozooplankton and therefore we described them within the mesozooplankton section. However, they showed completely different distribution patterns and relation to the environmental factors. So we had to consider them separately. This is really a bit of confusion. To avoid this, we described the results for polychaeta species in the text referring to Figure 10.

 Point 10: Page 17, Line 629: It seems like there is something missing in the beginning of this sentence.

Response 10: The sentences Line 629-630 “Its re.. from the Weddell Sea…” was corrected to “It’s replenishment from the Weddell Sea…”

Point 11: Regarding the conclusions, I believe most of what you have written, are more like a discussion. May be try to shorten a little bit the conclusions section and use some of the information in the discussion.

Response 11: Conclusions was combined with the discussion.

Reviewer 3 Report

The manuscript "Composition and Structure of Plankton Communities in the Atlantic Sector of the Antarctic under Conditions of Continuing Climate Warming" by Kasyan et al. contains rich information on diatoms, invertebrate and vertebrate (ichtyofauna) communities in the large study area (Atlantic Antarctic). I have included a file with small suggestions, other brief remarks are included below:

1. The title can be shortened (delete "Conditions of", as it could seem to imply experimental conditions mimicking warming (you are using natural data).

2. The reference to WoRMS (line 143) is not sufficient (it is not completely reliable for taxonomy), please add other taxonomical works you used for ID.

3. Some figures (Fig. 10, 13 for example) have very small letters and numbers, please increase the size (or font size).

4. The conclusions could be better structured. There is alot of data in the paper that is presented in the discussion. However the conclusions can be more brief focusing on a few major points in relation to the effects of climate change. Also, you mention in lines 636-637 (discussion) a hypothesis that should be explained more there, and that can be referred to again in conclusion. At present, I was a bit lost in the conclusions as to which major (3-5 max.) points are the main conclusions of this study.

Therefore I suggest "minor revisions" to the paper.

The use of English is excellent and does not need further detailed improvement

With kind Regards

Author Response

We are deeply thankful for consideration of the manuscript and valuable comments and suggestions made by reviewers that have greatly improved the manuscript. We provide point-by-point answers to comments of reviewers and highlighted changes in the manuscript using tracked changes. We hope that you will find our responses to reviewers’ comments satisfactory, and that the manuscript may now be acceptable for publication.

Response to Reviewer 3 Comments

Point 1: The title can be shortened (delete "Conditions of", as it could seem to imply experimental conditions mimicking warming (you are using natural data).

Response 1: We changed the title to “Composition and Distribution of Plankton Communities in the Atlantic Sector of the Southern Ocean”

Point 2: The reference to WoRMS (line 143) is not sufficient (it is not completely reliable for taxonomy), please add other taxonomical works you used for ID.

Response 2: The WoRMS was mainly used for correct name identification. Taxonomic literature sited after in each paragraph. Particular references added for phytoplankton to Ln 157, the reference for mesozooplankton to Ln 160, the reference for macrozooplankton to Ln 167, the reference for ichthyoplankton to Ln 171.

Point 3: Some figures (Fig. 10, 13 for example) have very small letters and numbers, please increase the size (or font size).

Response 3: We corrected figures 10, 13 and 14 with large size font.

Point 4: The conclusions could be better structured. There is a lot of data in the paper that is presented in the discussion. However the conclusions can be more brief focusing on a few major points in relation to the effects of climate change. Also, you mention in lines 636-637 (discussion) a hypothesis that should be explained more there, and that can be referred to again in conclusion. At present, I was a bit lost in the conclusions as to which major (3-5 max.) points are the main conclusions of this study.

Response 4: Conclusions was combined with the discussion.

 Also, the entire text has been corrected according to Reviewer 3 remarks and suggestions.

Round 2

Reviewer 1 Report

Authors marginally improved MS but in my view failed to improve discussion by linking different parts the plankton community together. Also, extensive literature on the topic accumulated in this region (perhaps the best studied region of the Southern Ocean) is very poorly utilized. If this this the best authors can provide, I think nothing else can be done. The MS does include novel data but has no proper discussion placing it in the contest of the region oceanographic setting and historical comparisons. As a report, it can be published. However, the data set seems to be heavily underutilized.

Lastly, there is still no explanation from which depth mesozooplankton data were analyzed, top 200 m or top 500 m.

Author Response

We are sincerely grateful to the reviewers for their efforts and valuable recommendations and agree with most of reviewers’ comments.

Responses to Reviewer’s 1 Comments 2

Point 1: Authors marginally improved MS but in my view failed to improve discussion by linking different parts the plankton community together.

 Response 1: Possibly, there is not enough discussion in our draft regarding links between different parts of the plankton community, nonetheless accumulation of basic information is very important and necessary to detect fundamental long-term climate changes scenarios.

We added few changes in discussion/conclusion section to improve the text.

Point 2: Also, extensive literature on the topic accumulated in this region (perhaps the best studied region of the Southern Ocean) is very poorly utilized. If this the best authors can provide, I think nothing else can be done.

 Response 2:

The list of literature used in the manuscript is quite large and we are not in favour to expand it even more. In the discussion part, we tried to use the most up-to-date information. However, we added some references.

  1. Hill S.L., Phillips T., Atkinson A. Potential Climate Change Effects on the Habitat of Antarctic Krill in the Weddell Quadrant of the Southern Ocean. PLoS ONE. 2013. 8(8): e72246. https://doi.org/10.1371/journal.pone.0072246
  2. Murphy, E.J.; Johnston, N.M; Hofmann, E.E.; Phillips, R.A.; Jackson, J.A.; Constable, A.J.; Henley, S.F.; Melbourne-Thomas, J.; Trebilco, R.; Cavanagh, R.D. et al. Global Connectivity of Southern Ocean Ecosystems. Frontiers in Ecology Evolution. 2021. 9:624451. doi: 10.3389/fevo.2021.624451
  3. McCormack, S.A.; Melbourne-Thomas, J.; Trebilco, R.; Griffith, G.; Hill, S.L.; Hoover, C.; Johnston, N.M.; Marina, T.I.; Murphy, E.J.; Pakhomov, E.A. et al. Southern Ocean Food Web Modelling: Progress, Prognoses, and Future Priorities for Research and Policy Makers. Frontiers in Ecology Evolution. 2021. 9:624763. doi: 10.3389/fevo.2021.624763

Point 3: The MS does include novel data but has no proper discussion placing it in the contest of the region oceanographic setting and historical comparisons. As a report, it can be published. However, the data set seems to be heavily underutilized.

Response 3:

In discussion/conclusion section we added the sentences “Aggregations of phytoplankton (Actinocyclus, Asteromphalus and Fragilariopsis), mesozooplankton, pelagic polychaetes, euphausiids T. macrura and tunicate S. thompsoni found in the waters of the South Orkney Islands with high values water temperatures and with low values of salinity were more abundant than these groups in the shelf waters off the Antarctic Peninsula. Whereas the distribution of phytoplankton (Chaetoceros, Odontella, Phaeocystis and Corethron), krill and larvae of fish mainly located in the waters off the Antarctic Peninsula with low temperatures, high values of salinity and Chl a concentration”. Previous studies have reported a substantial increases in Chl a occurring farther south of 63°S [167]. Such changes are associated with changes in the composition of the phytoplankton community and consistent with the expected widespread consequences of marine warming [167,168]. We observed a significant reorganization of the structure of phytoplankton communities and a decrease some mass of diatoms relative to other phytoplankton. A reduction in diatoms in the diet of Antarctic krill is likely to reduce both growth and reproduction [97,126,127].”

Concerning the pelagic polychaete taxocene, our findings show that this group is distributed in a different way than others and analyses performed showed no significant relation with any other variable except for the sea ice cover.

In discussion/conclusion section we added the sentences “There are no distinct spatial groups or taxocenes of pelagic polychaetes in the study area. The most widespread species are omnivores with a predominance of phytoplankton in their diet. However, their distribution was negatively correlated with the abundance of phytoplankton, as it was instead determined by the composition of the predominant microalgae species. Carnivorous species’ abundance, on the contrary, followed the trend in abundance of their prey, showing no relation with its species composition. This is expectable taking into consideration that the observed carnivorous polychaete species consume any prey with a suitable body size. As for the most specialized group, Typhloscolecidae, no preference for certain species of chaetognaths has been shown so far, and high abundance of the latter group typical for all stations with the later succession stage brought to higher abundances of typhloscolecids. Thus, the observed differences between stations followed the overall seasonal succession the best proxy for which is the time of the waters freeing from closed ice”.

Point 4: Lastly, there is still no explanation from which depth mesozooplankton data were analyzed, top 200 m or top 500 m.

Response 4: Upper 500 m layer at the deep-sea stations (by Multinet) and upper 200 m layer at the non deep-sea stations (by WP-2 net) was discussed in detail in the manuscript. Abundance for mesozooplankton taxa was determined from each nets sample. A total of more 90 mesozooplankton samples were analyzed.

Thank you very much for your comment and valuable opinion. We would be grateful for the Reviewer to have the improved manuscript as it is. The aim of the study was to present the results of studying the taxonomic composition and quantitative distribution of plankton communities in Bransfield Strait, Antarctic Sound, the Powell Basin of the Weddell Sea and the waters off the Antarctic Peninsula and South Orkney Islands during the austral summer of 2022. Among of the obtained results, we have one certain and clear one "the studied macrozooplankton species are tied-in to a certain combination of hydrophysical factors, among which the major abiotic factor was the surface temperature, and the major biotic one was the availability of forage resources". We discussed the results and our statement in MS, and we refer to our first findings (a reference to Murzina et al., 2022). More, we are deeply discussed these in the preparing MS to Water (MDPI): Murzina S.A., Voronin V.P., Bituitskii D.G., Pekkoeva S.N., Frey D.I., Orlov A.M. Fatty acid spectrum of Antarctic krill Euphausia superba indicates the hydrophysical and trophic conditions of environment // Water. Special Issue "Physical and Biological Properties of Waters in the Region of the Antarctic Peninsula and Adjacent Basins of the South Atlantic". This main idea will be revealed in the MS.

Reviewer 2 Report

diversity-1944138R2

The manuscript has been revised following the reviewers’ suggestions and answering the questions posed to clarify some aspects. I appreciate the changes that the authors have made throughout the manuscript. It was a huge improve and I believe the manuscript is almost ready for publication. I still have a final suggestion:

- In the Materials and Methods section, maybe is better to put the first sentence of 2.2 Sampling: “Data were collected during the 87th cruise of the R/V Akademik Mstislav Keldysh in January-February 2022. Research areas included the Bransfield Straits, the Antarctic Sound, the Powell Basin of the Weddell Sea, the waters off the Antarctic Peninsula and the South Orkney Islands (Figure 1).” at the beginning of 2.1 Oceanographic measurements.

- Also, I would change the name of the subsection to “2.2 Biological sampling”.

Author Response

We agree with most of reviewer’s comments. We are sincerely grateful to the reviewers for their efforts and valuable recommendations.

Response to Reviewer 2 Comments 2

Point 1: In the Materials and Methods section, may be is better to put the first sentence of 2.2 Sampling: “Data were collected during the 87th cruise of the R/V Akademik Mstislav Keldysh in January-February 2022. Research areas included the Bransfield Straits, the Antarctic Sound, the Powell Basin of the Weddell Sea, the waters off the Antarctic Peninsula and the South Orkney Islands (Figure 1)” at the beginning of 2.1 Oceanographic measurements.

Response 2: Done

Point 2: Also, I would change the name of the subsection to “2.2 Biological sampling”.

Response 2: Done
